# Redistribution of PU.1 partner transcription factor RUNX1 binding secures cell survival during leukemogenesis

Alexander Bender[1,5], Füsun Boydere[1,5], Ashok Kumar Jayavelu[2], Alessia Tibello[1], Thorsten König[1], Hanna Aleth[1], Gerd Meyer zu Hörste[3], Thomas Vogl[4] & Frank Rosenbauer [1✉]

## Abstract

**Transcription factors (TFs) orchestrating lineage-development often control genes required for cellular survival. However, it is not well understood how cells survive when such TFs are lost, for example in cancer. PU.1 is an essential TF for myeloid fate, and mice with downregulated PU.1 levels develop acute myeloid leukemia (AML). Combining a multi-omics approach with a functional genetic screen, we reveal that PU.1-downregulated cells fundamentally change their survival control from cytokine-driven pathways to overexpression of an autophagy-predominated stem cell gene program, for which we also find evidence in human AML. Control of this program involves redirected chromatin occupancy of the PU.1 partner TF Runx1 to a lineage-inappropriate binding site repertoire. Hence, genomic reallocation of TF binding upon loss of a partner TF can act as a pro-oncogenic failsafe mechanism by sustaining cell survival during leukemogenesis.**

**Keywords** PU.1; RUNX1; Myeloid Development; Myeloid Leukemia
**Subject Categories** Cancer; Chromatin, Transcription & Genomics

## Introduction

Cell lineage development requires precise and temporally resolved regulation of gene expression orchestrated by transcription factors (TFs). However, in addition to inducing genes that drive the differentiation process and establish cellular lineage identity, the same TFs can also control genes essential for cellular survival and proliferation, such as receptors for cytokines and growth factors (Rauch and Mandrup, 2021). Hence, this raises the question of how progenitor cells can survive in cases when TFs are deactivated or downregulated, for example, in cancerogenesis. This is important, because cancerogenesis is a multi-step process, during which several or even multiple mutations must be acquired to complete malignant transformation (Vogelstein and Kinzler, 1993). Hence, the formation of cancer cells usually requires many years, during which the precancerous precursors must persist until the transformation process is complete.

A central TF involved in controlling hematopoietic precursors, especially of the myeloid lineage, is PU.1 (Rosenbauer and Tenen, 2007). PU.1 knockout mice cannot generate monocytes, macrophages, or B cells (Scott et al, 1994; McKercher et al, 1996) but do produce neutrophils (Dakic et al, 2005). Further, as occurs in T-cell development, where PU.1 is required for the production of early progenitors but is downregulated before they commit to the T lineage (Hosokawa et al, 2021), dosage-dependent up- and downregulation of PU.1 expression is key for the development of all hematopoietic lineages (Nutt et al, 2005).

In our previous studies with mice lacking a key *Spi1* (encoding PU.1) *cis*-enhancer termed upstream regulatory element (URE), we found that URE[Δ] animals, which harbor downregulated PU.1 expression, eventually succumb to lethal acute myeloid leukemia (AML) (Rosenbauer et al, 2004). Prior to AML, all mutant animals undergo a months-long preleukemic phase during which the involved cells acquire further mutations. However, PU.1 deficiency compromises hematopoietic stem cell (HSC) functions and produces myeloid progenitors expressing a limited repertoire of growth factor receptors (DeKoter et al, 1998; Iwasaki et al, 2005; Carotta et al, 2010), raising the question of how preleukemic URE[Δ] cells can persist until their full malignant transformation is completed. This question is important because PU.1 expression or activity is often impaired in AML, so that understanding how myeloid progenitors can survive PU.1 downregulation is of potential translational interest (Antony-Debré et al, 2017; Takei and Kobayashi, 2019). In this report, we show that PU.1-downregulated cells maintain their survival by overexpressing a RUNX1-dependent stem cell-derived growth program predominated by autophagy.

[1]Institute of Molecular Tumor Biology, University of Münster, Münster, Germany. [2]Proteomics and Cancer Cell Signaling, Clinical Cooperation Unit Pediatric Leukemia, German Cancer Research Center (DKFZ) and Hopps Children's Cancer Center (KiTZ), University of Heidelberg, Heidelberg, Germany. [3]Department of Neurology with Institute of Translational Neurology, University of Münster, Münster, Germany. [4]Institute of Immunology, University of Münster, Münster, Germany. [5]These authors contributed equally: Alexander Bender, Füsun Boydere. ✉E-mail: frank.rosenbauer@ukmuenster.de

# Results

## PU.1 downregulation reverses myeloid lineage fates early in HSCs

We first assessed lineage-negative c-Kit[+] bone marrow cells (containing HSPCs) from preleukemic URE[Δ] and corresponding wild-type (WT) mice by single-cell (sc) RNA-seq (Appendix Fig. S1A). After quality control (Appendix Fig. S1B–E) and removal of dendritic cells, we obtained eight clusters, which we annotated based on established marker genes (Paul et al, 2015; Nestorowa et al, 2016; Giladi et al, 2018) (Fig. 1A,B; Appendix Fig. S1F,G; Dataset EV1). We observed HSCs and early progenitors (*Hoxa9, Hlf, Flt3, Ly6a*) in cluster C1, neutrophil progenitors (*Elane, S100a8, S100a9*) in C3 and monocyte progenitors (*Ly86, Csf1r, Irf8*) in C5. C2 expressed markers of several clusters at moderate levels, suggesting a developmentally intermediate progenitor stage. Interestingly, C1 expressed only 2.3% of the PU.1 levels detected in WT counterparts which was the largest reduction across all clusters, suggesting that PU.1 expression in early precursor cells relied most on URE enhancer activity (Figs. 1C and S1H,I). Furthermore, C3 was notably enriched in URE[Δ] mice, while C5 was depleted (Figs. 1D and S1J; Table EV1), in line with the neutrophilic and monocytopenic phenotype of URE[Δ] mice (Rosenbauer et al, 2004). C4 (which almost exclusively consisted of URE[Δ] cells), was predominated by neutrophil genes (Dataset EV1), and was in accordance with the recently described preNeu neutrophil progenitor (Evrard et al, 2018). Remaining clusters C6 was annotated as basophil progenitors (*Prss34, Lmo4,* and *Gata2*), C7 and C8 were assigned megakaryocyte and erythroid progenitors (e.g., *Pf4* and *Car1*), respectively.

Next, we calculated neutrophil and monocyte lineage scores for C1 and C2 cells which were located developmentally upstream of neutrophil (C3) and monocyte (C5) progenitors (Fig. 1E). These scores supported that C1 cells already show a general transcriptional shift toward a neutrophil identity at the cost of downregulated monocyte identity. We then visualized the expression of these neutrophil and monocyte lineage signatures along the pseudotime trajectory (Fig. 1F). URE[Δ] mice overexpressed the neutrophil gene signature even in the most upstream C1 cells while failing to induce the monocyte gene signature.

Taken together, we found that URE[Δ] HSCs, including the most upstream ones, overexpressed neutrophil genes while underexpressing monocytic genes. Hence, PU.1 downregulation reverses myeloid lineage choice in the most immature HSCs, thus connecting prematurely enforced neutrophil differentiation with stem cell features.

## PU.1 controls growth and differentiation programs simultaneously in the same cells

We next tracked transcriptional changes during HSPC differentiation along the neutrophil trajectory (clusters C1, C2, and C3). We identified differentially expressed genes between URE[Δ] and WT cells of these clusters using pseudobulk analysis resulting in a total of 1919 genes. Hierarchical clustering of these genes in clusters C1, C2, C3, and C5 (the latter to capture monocyte differentiation genes already primed in early HSPCs) yielded four distinct groups, hereafter termed signatures (Fig. 2A,B;

Dataset EV2). Of note, pseudotime tracking showed that these signatures formed a dynamic continuum of up- and downregulated genes along the HSPC-derived progenitor trajectory (Fig. 2C).

Signatures 1 (270 genes) and 2 (591 genes) were underexpressed in URE[Δ] cells (Fig. 2A,B—upper panels). While signature 1 genes were most highly expressed in C1 cells and were reduced towards C3, signature 2 genes followed the reverse path. Signature 1 included bona fide stem cell markers (*Cd34, Hoxa9, Meis1*), cytokine receptors (*Il1r1, Il6st, Flt3*), as well as cell cycle (*Ccnd1/2*), proliferation (*Mki67*), and apoptosis (*Casp3*) regulators, and was enriched for "JAK-STAT", "AGE-RAGE" and "Hedgehog" signaling terms. Hence, signature 1 comprised PU.1-activated stem cells and early progenitor genes involved in growth signaling and proliferation.

Signature 2 included the PU.1 gene (*Spi1*) itself as well as well-known PU.1-controlled cytokine receptors (*Csf1r, Csf2ra*) and monocyte markers (*Ly86, Ccr2*), which together were part of the enriched pathway term "Cytokine-cytokine receptor interactions", suggesting a loss in a PU.1-activated myeloid differentiation and growth program.

Signatures 3 (346 genes) and 4 (712 genes) were overexpressed in URE[Δ] cells (Fig. 2A,B—lower panels). Signature 3 was most highly expressed in URE[Δ] C1, and comprised many signaling factor-encoding genes, altogether converging in pathway enrichments of Rap1 signaling, calcium signaling, and PI3K-Akt signaling pathways. It also contained the anti-apoptosis gene *Bcl2*. Hence, signature 3 represented genes with potential relevance for HSPC survival. Signature 4 showed a continuous increase from C1 to C3 in both genotypes, but with noticeably higher levels in the URE[Δ] clusters. Functional annotation suggested that this signature was dominated by neutrophil genes, and, as such, was most likely the major driver of the neutrophilia observed in URE[Δ] mice.

Collectively, URE[Δ] cells differentially regulated two distinct stem cell programs, each predominated by different survival genes, suggesting a growth regulation that is fundamentally distinct from that of WT cells. Moreover, URE[Δ] cells underwent deregulation of both stem cell and myeloid genes simultaneously in the same cells, suggesting that the control of cell survival and differentiation is interconnected by PU.1.

## Transcriptional changes in URE[Δ] cells are translated to the proteome

We next undertook a proteomics approach to address whether the transcriptomic changes seen in URE[Δ] cells are translated into global protein changes. To collect enough protein from a homogenous cell population, we first generated non-transformed myeloid progenitor cell lines from preleukemic URE[Δ] and WT mice by immortalizing lineage-negative c-Kit[+] cells of the bone marrow with a Hoxb8-ER fusion construct (Wang et al, 2006). Hoxb8-ER[+] URE[Δ] cells (hereafter termed "Hox-URE" cells) demonstrated greatly reduced PU.1 protein as expected (Appendix Fig. S2A). Both lines Hox-URE and Hox-WT expressed c-Kit but not Sca-1, highlighting their progenitor identity, which we confirmed by cell morphology (Appendix Fig. S2B,C). Bulk RNA-seq from these cell lines followed by gene set enrichment analysis (GSEA) confirmed that the expression patterns of the aforementioned gene signatures were preserved (Appendix Fig. S2D). Label-free mass spectrometry

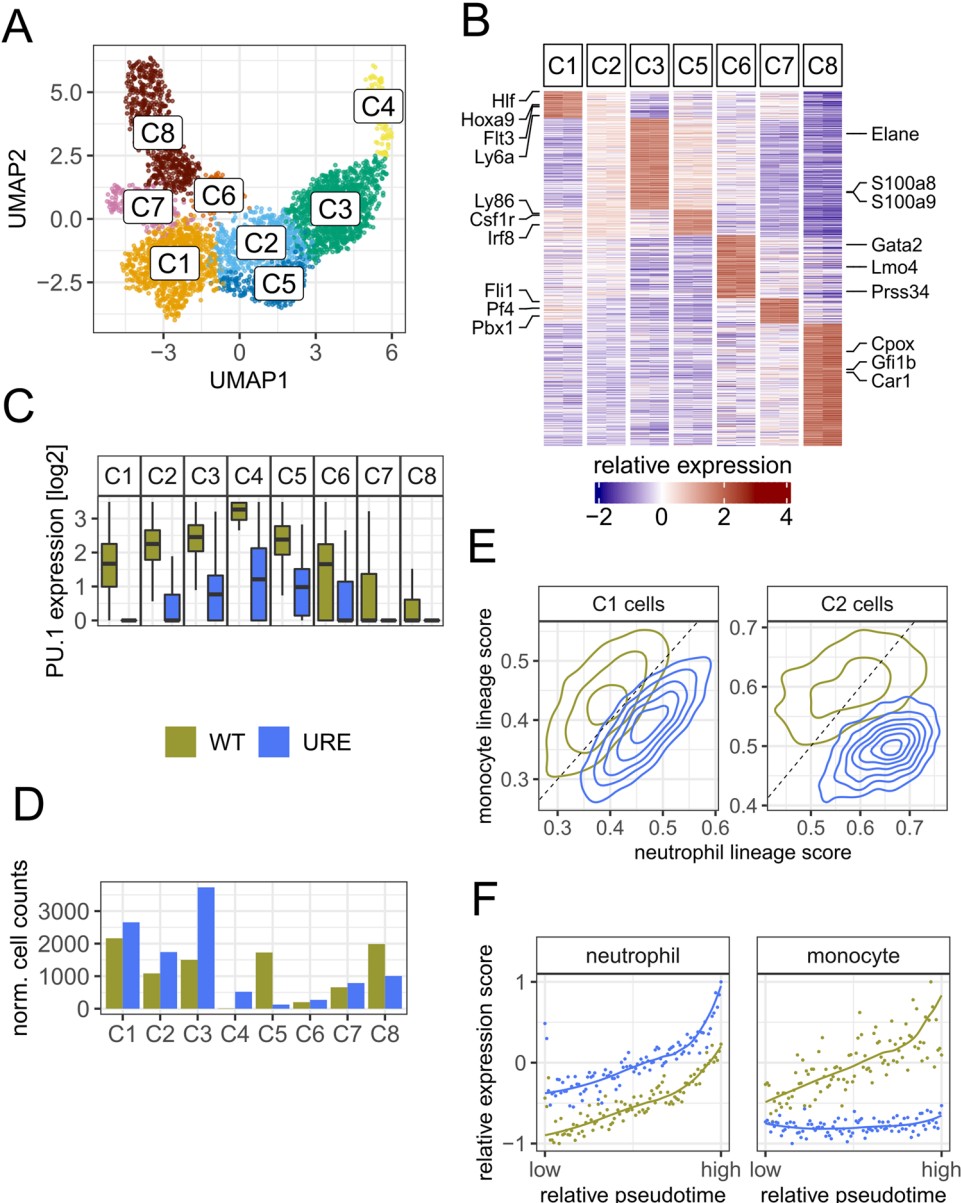

**Figure 1. Myeloid lineage conversion in most upstream HSCs upon PU.1 downregulation.**

(A) UMAP representation of the integrated dataset colored by clusters. (B) Heatmap summarizing relative expression of marker genes per cluster based on WT cells. C4 cells were excluded due to low cell counts in WT samples. (C) Gene expression of PU.1 (*Spi1*) per cluster (single-cell resolution, *n* ~ 10,000 cells in total). The horizontal line represented the median, upper and lower hinges represented the 25th and 75th quartile, whiskers represent data no more distant from the whiskers than 1.5 times the interquartile range, outliers beyond the whiskery were not shown. (D) Number of cells per cluster, normalized to 10,000 cells per genotype. (E) Contour plots of lineage progenitor scores for cells in clusters C1 and C2. For the neutrophil lineage, we used C3 marker genes, while for the monocyte lineage, we used C5 marker genes. (F) Relative expression of the neutrophil and monocyte lineage genes for C1 cells ordered by pseudotime along an estimated trajectory.

assessment of protein levels within the cells (Cox et al, 2014) detected 5616 unique proteins, of which 957 were increased, and 604 (including PU.1) were decreased in Hox-URE as compared to Hox-WT cells, respectively (Fig. 2D). Importantly, GSEA demonstrated that the deregulation of all four transcriptional signatures in URE$^\Delta$ cells was replicated in the proteome (Fig. 2E). This suggests global biological relevance of the deregulated gene signatures and highlights the suitability of the Hox cell lines for studies on the URE$^\Delta$ phenotype.

## PU.1 downregulation overactivates a comprehensive pro-survival gene program

Stem cells are canonically characterized by their ability to self-renew and survive stress conditions, which are important features conferred to leukemic cells. Signature 3 was most interesting in this context, because it was most highly expressed in early URE$^\Delta$ HSPCs which formed the origin of AML. Therefore, we addressed whether overexpression of this signature was the basis for the growth of

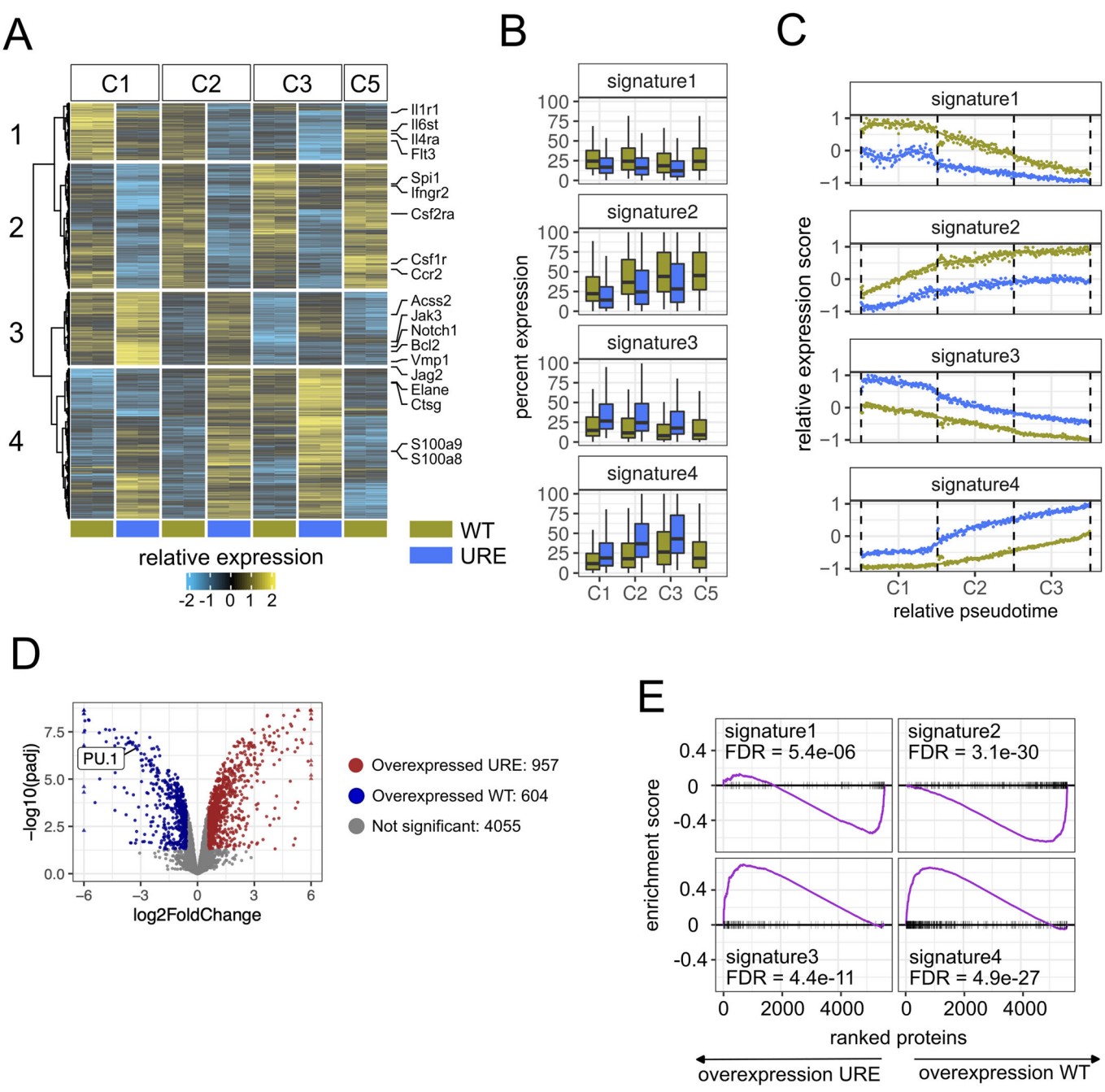

**Figure 2. Combined conversion of myeloid lineage and stem cell growth programs in PU.1-downregulated cells.**

(A) Heatmap representing the relative expression of differential genes (R/Bioconductor package DESeq2, FDR < 0.005, absolute fold change >1.5, pseudobulk aggregation per each donor mouse, $n = 2$ mice per condition) between URE vs WT cells of cluster C1, C2, and C3. Positive expression values indicate overexpression in URE cells. C5 for URE cells was excluded as this cluster was notably depleted in this genotype (see Fig. 1D). (B) Boxplots representing the percentage of expression per cluster towards signature genes. A gene was considered expressed in the case of non-zero counts (single-cell resolution, $n$ ~10,000 cells in total). Horizontal line represented the median, upper and lower hinges represented the 25th and 75th quartile, whiskers represent data no more distant from the whiskers than 1.5 times the interquartile range, outliers beyond the whiskery were not shown. (C) Average expression score of the four gene signatures from (A) visualized along a neutrophil lineage pseudotime trajectory starting in cluster C1 and ending in cluster C3. (D) Volcano plot of differential protein expression analysis in Hox-URE vs Hox-WT cells (FDR < 0.05 from a linear model using the R/Bioconductor package limma, minimum absolute fold change 1.5, $n = 4$ cell line replicates per condition). (E) Gene set enrichment analysis using proteome data from Hox-URE vs Hox-WT data. FDRs were calculated using a permutation-based test in the R/Bioconductor package fgsea.

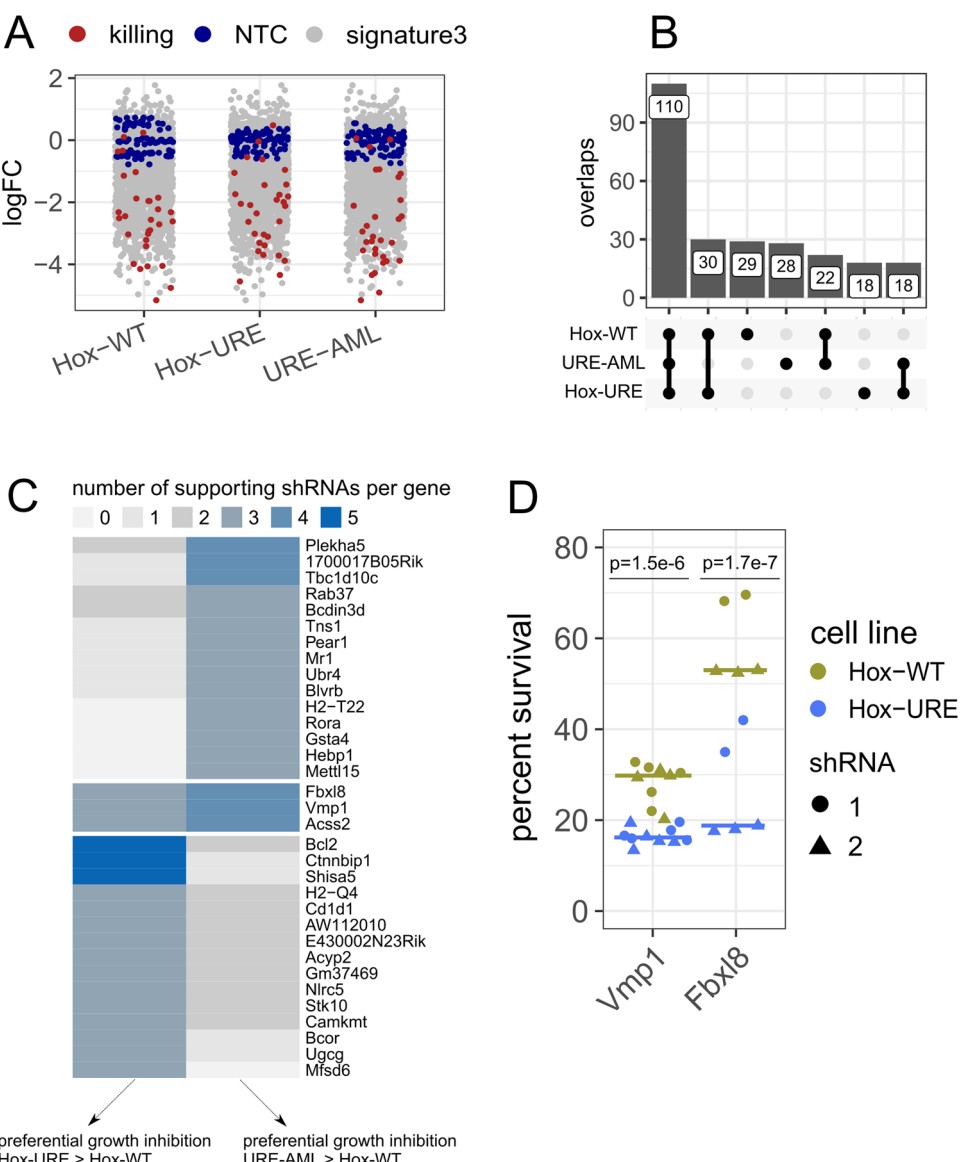

**Figure 3. Large-scale remodeling of the cell survival program in PU.1-downregulated cells.**

(A) Quantile-normalized fold changes of individual shRNAs comparing selected libraries (harvested after 10 cell divisions, see Appendix Fig. S3A) to its input controls. Killing (red) and non-targeting (NTC, blue) controls were highlighted. Target genes of signature 3 are shown in gray. (B) Upset plot summarizing overlaps between the sets of essential genes per cell line. (C) Heatmap summarizing genes identified to show preferential growth inhibition in Hox-URE (left) or URE-AML (right) compared to Hox-WT based on differential logFCs (see Methods). Three or more shRNAs per gene were required to support this classification. (D) Low-throughput validation experiment towards preferential growth inhibition after knockdown of indicated genes with two independent shRNAs in Hox-WT and Hox-URE cells. Survival was measured four days after transduction. P values were calculated by one-way ANOVA followed by multiple testing correction (Benjamini–Hochberg) comparing the percent survival between cell lines (n = 5–10). Source data are available online for this figure.

URE^Δ cells. To this end, we performed a focused shRNA screen covering all signature 3 genes in Hox-WT and Hox-URE cells and assayed cell growth (Appendix Fig. S3A). To also cover the leukemic state, we established an AML cell line (termed URE-AML) from leukemic blasts of a diseased URE^Δ mouse (Appendix Fig. S2A–C), which we included in the screen.

We designed a pooled library with five individual shRNAs for each of the signature 3 genes as well as a total of 30 positive/killing (*Psma1, Rpl30, Polr2b*) and 45 non-targeting (NTC) luciferase controls for both normalization and control of background noise.

We lentivirally delivered the shRNA constructs into the cell lines and allowed them to grow for 10 cell divisions. As a baseline control, we harvested cells 24 h after library transfection. Assuming that the shRNA-mediated knockdown of a gene would cause reduced cellular growth, the frequency of the shRNA-attached barcodes would decrease in the total population. Relative representation of each shRNA barcode (both input control libraries and final libraries after 10 cell divisions) was quantified by sequencing the shRNA-attached barcodes (see Methods and Appendix Methods).

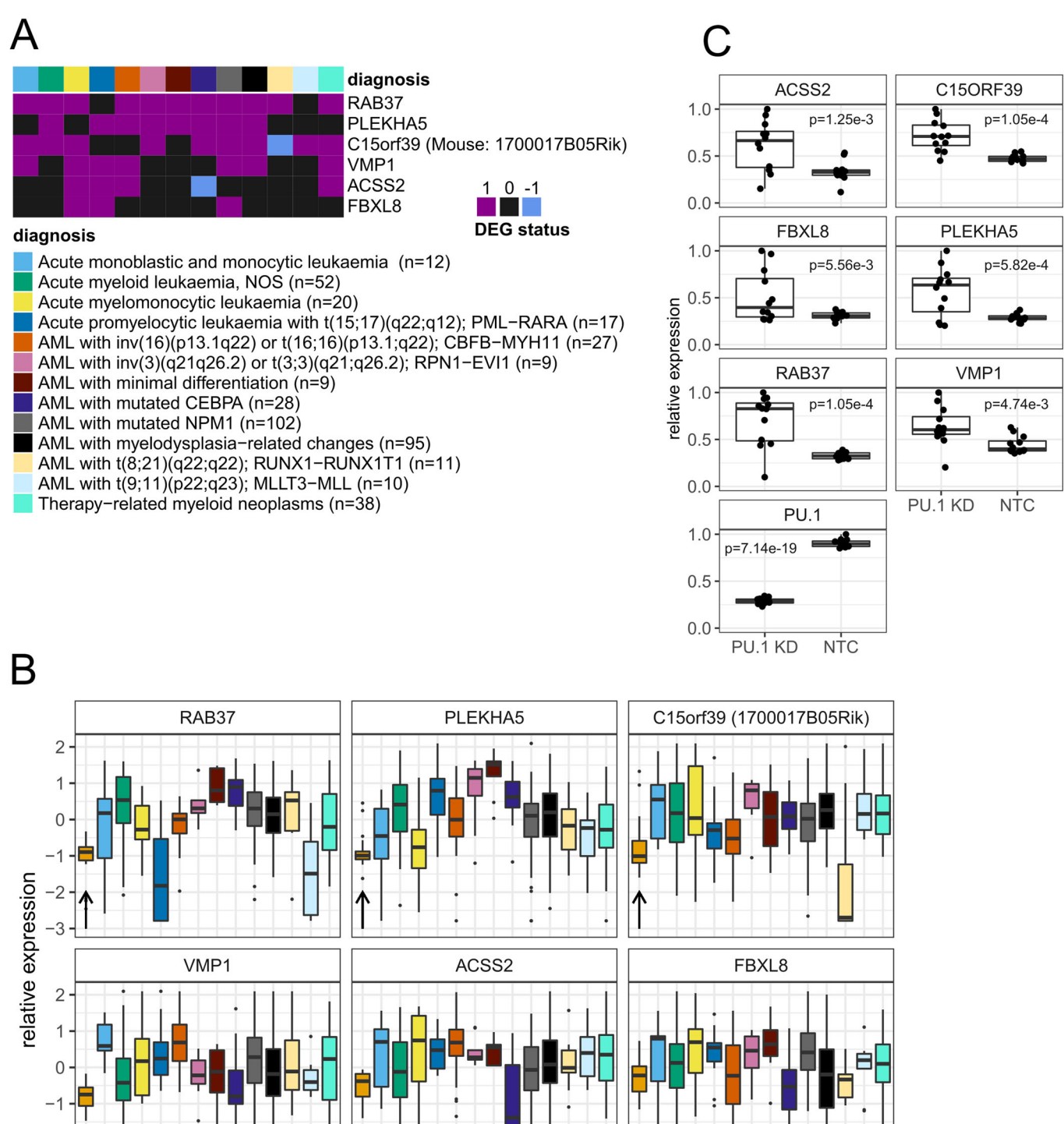

We did not find evidence for a transduction bias (Appendix Fig. S3B). Furthermore, the killing control shRNAs caused a strong reduction in cellular survival and separated well from the NTC population (Fig. 3A). We next used MAGeCK (Li et al, 2014) to identify "essential" genes per cell line, defined as those that (a) caused reduced growth upon knockdown, (b) were statistically significant in a permutation test (FDR < 0.01), and (c) were supported by a significant knockdown by at least three of the five shRNAs to minimize the influence of off-target effects and experimental outliers. Congruently, we obtained 191, 176 and 178 essential genes in Hox-WT, Hox-URE, and URE-AML cell lines, respectively (Dataset EV3). This high number of essential genes functionally revealed that a large part of signature 3, which was overexpressed in URE$^\Delta$ cells, consisted of genes promoting growth and survival.

**Figure 4.  Certain PU.1-controlled survival genes are often upregulated in human AML.**

(**A**) RNA-seq data from the BeatAML cohort (Tyner et al, 2018) containing several AML subtypes and healthy control samples (19 mononuclear cell samples and two CD34+ samples) were tested for differential expression (FDR < 0.01, absolute FC > 1.5, R/Bioconductor package limma) and then colored based on their differential expression status (1 = overexpressed, 0 = not significant, −1 = underexpressed in AML vs control. (**B**) Boxplot representation of relative expression values of the genes in (**A**) that were differentially expressed in three or more comparisons. The color code is identical to (**A**), and the control group is colored in orange, highlighted with an arrow. Sample sizes were indicated in the legend of panel (**A**). (**C**) Gene expression changes (qPCR) in THP-1 cells after PU.1 knockdown with a PU.1-specific shRNA or non-targeting control (NTC). Gene expression values were displayed relative to the sample with the highest expression per gene, which was set to 1. This was the input for the calculation of $p$ values using one-way ANOVA and correction for multiple testing by Benjamini–Hochberg ($n = 12$, independent RNA extractions from cell line isogenic replicates). For (**B, C**), the horizontal line represented the median, upper and lower hinges represented the 25th and 75th quartile, whiskers represent data no more distant from the whiskers than 1.5 times the interquartile range, outliers beyond the whiskery were not shown. In (**C**), all data points were additionally shown as dots. Source data are available online for this figure.

## PU.1 decrease leads to overexpression of pro-survival genes

Genes of potential therapeutic relevance in AML are those that, upon inhibition, reduce the growth and survival of leukemic cells more than of healthy cells. Most (110) of the essential signature 3 genes were found to be essential in all three cell lines, highlighting their broad relevance for cell growth (Fig. 3B). We expanded our analysis to quantitatively identify those essential genes that, upon knockdown, showed preferential growth inhibition in preleukemic and leukemic cells compared to WT controls. For this, we called genes as essential (according to the definition given above) in URE-AML and Hox-URE and then calculated differential fold changes (DFC) towards Hox-WT cells. We then selected those genes with negative DFCs (meaning preferential growth inhibition in URE-AML or Hox-URE vs WT, respectively) beyond an empirical cutoff based on DFCs of the NTCs (see Methods and Appendix Methods). This analysis yielded 18 genes for Hox-URE cells and also 18 genes for URE-AML cells with preferential growth inhibition in these cells compared to the WT control, respectively (Figs. 3C and S3C,D).

Notably, three of these genes (*Fbxl8, Vmp1, Acss2*) were classified to show preferential growth inhibition in both Hox-URE and URE-AML than Hox-WT cells, suggesting their importance for both URE$^\Delta$ preleukemic and leukemic states. Fbxl8 is an E3 ubiquitin ligase with oncogenic functions in breast cancer (Chang et al, 2020) but tumor suppressor functions in lymphoma (Yoshida et al, 2021). Vmp1 is a key player in autophagy induction (Ropolo et al, 2007; Wang et al, 2020), and nucleus-translocated Acss2 has been shown to regulate the transcription of autophagy-related genes (Li et al, 2017). We confirmed shRNA knockdown by qPCR and validated the pro-survival function of these genes by transducing URE-AML cells individually with shRNAs (inserted into a lentiviral GFP reporter plasmid) followed by tracing GFP levels over time (Appendix Fig. S3E,F). While knockdown of Acss2 showed significant but small retardation of cell survival, knockdown of either Vmp1 or Fbxl8 had much greater effects, matching that of the killing control. Therefore, we compared the effects of Vmp1 or Fbxl8 knockdown in Hox-URE versus Hox-WT cells, confirming the preferential growth inhibition in PU.1-downregulated cells (Fig. 3D).

Collectively, the shRNA screen revealed that signature 3 comprised a high number of genes with essential roles for cell growth and survival. Importantly, several of these genes were classified to show preferential growth inhibition in URE$^\Delta$ than WT cells. We conclude that PU.1 downregulation promotes AML development by overexpressing stem cell-based survival genes.

## Human AML overactivates autophagy-linked genes that are permanently essential for URE$^\Delta$ leukemia

We next compared our data to human AML transcriptomes, focusing on 16 of the 18 signature 3 genes that showed evidence for preferential growth inhibition in URE-AML than WT cells, and for which there was a human ortholog (Table EV2). We assessed the expression status of these genes in the large human BeatAML cohort containing RNA-seq profiles of primary AMLs and healthy bone marrow (mononuclear and CD34-enriched cells) controls (Tyner et al, 2018). We compared AML subtypes with the healthy control groups and found six genes to be overexpressed in three or more subtypes, including those with known blocked PU.1 activity (e.g., t(8;21) and t(15;17)) (Fig. 4A,B). The overexpressed genes included the aforementioned *FBXL8*, *VMP1*, and *ACSS2*, which were identified as essential in both Hox-URE and URE-AML but not Hox-WT cells in our shRNA screen. Hence, human AML preferentially overexpressed PU.1-repressed genes within signature 3 that were required for both preleukemic and leukemic phases in the URE$^\Delta$ model. Supporting this notion, the genes overexpressed in human AML also contained *RAB37* and *PLEKHA5*, both of which were also considered essential for both Hox-URE and URE-AML but not Hox-WT cells by two shRNAs each. Knockdown of PU.1 by a tetracycline-inducible shRNA in THP-1 cells (Schuetzmann et al, 2018) led to enhanced expression of the aforementioned genes, confirming that these genes were downstream by PU.1 in human AML cells (Fig. 4C; Appendix Fig. S4A).

Notably, at least four (possibly all) of the six signature 3 genes overexpressed in human AML (*VMP1, ACSS2, RAB37*, and *PLEKHA5*) were directly or potentially related to cellular autophagy: besides the well-established role of VMP1 in generating autophagosomes, ACSS2 and RAB37 have also been reported as regulators of autophagy (Ropolo et al, 2007; Li et al, 2017; Sheng et al, 2018; Song et al, 2018; Wang et al, 2020). Moreover, autophagy was recently demonstrated to be regulated by cellular levels of the trace element copper (Tsang et al, 2020). PLEKHA5 controls copper homeostasis (Sluysmans et al, 2021), thus potentially linking PLEKHA5 to autophagy. Further, close ties exist between autophagy and the ubiquitin-proteasome system, indicating that FBXL8 may also be involved in autophagy (Chen et al, 2019).

## The proteome of PU.1-downregulated cells is enriched for autophagy-related pathways

We next asked whether we could find supporting evidence for increased autophagy upon PU.1 downregulation, using our

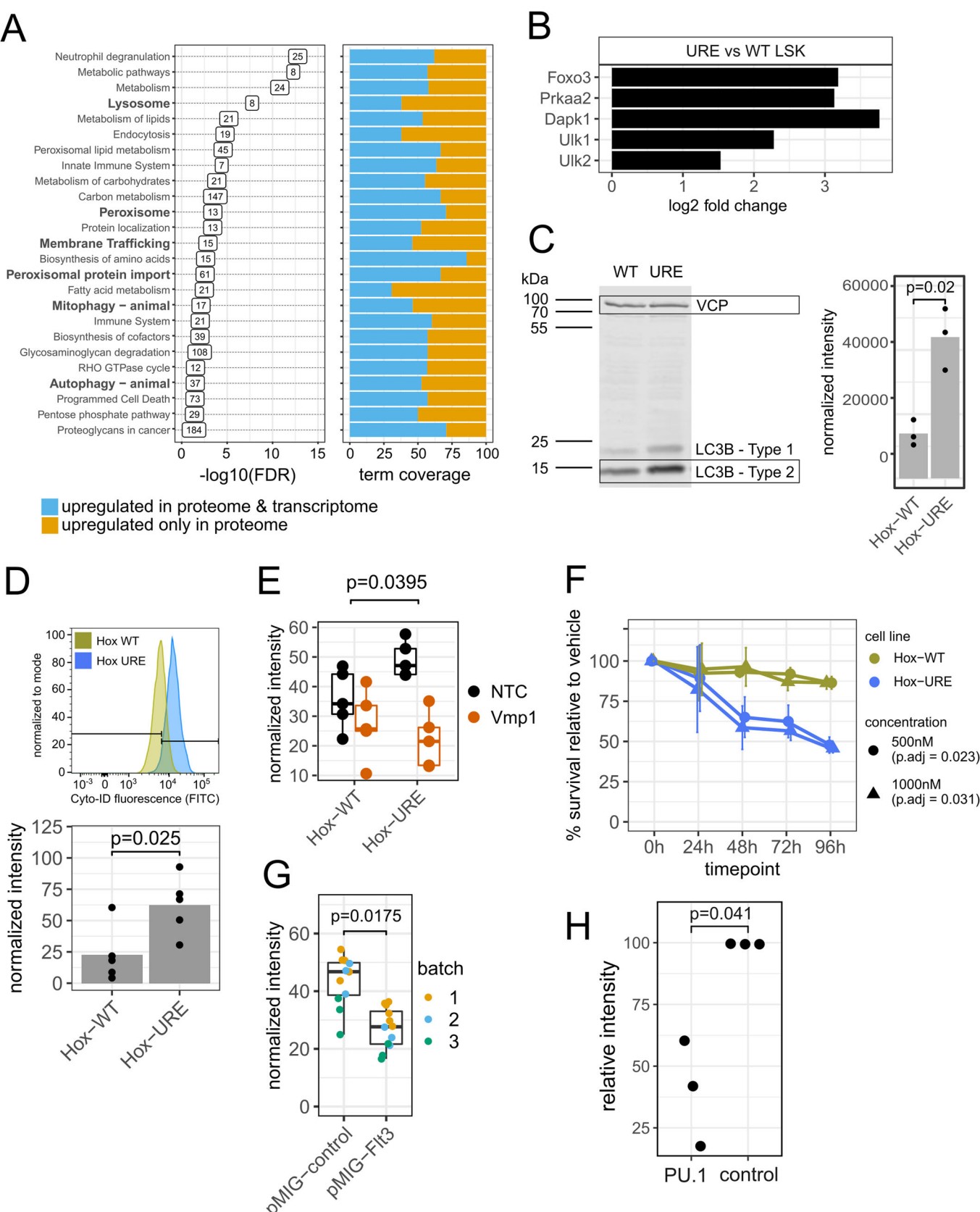

**Figure 5.  PU.1 downregulation reprograms cell survival towards autophagy.**

(A) Results from functional enrichment analysis of differential protein analysis. Protein levels per cell line were assessed by mass spectrometry (see Fig. 2D). Proteins overexpressed in Hox-URE versus Hox-WT cells (FDR < 0.05, absolute fold change >1.5, R/Bioconductor package limma) were analyzed for enrichments against KEGG and REACTOME databases with an FDR cutoff of 0.05 ($n = 4$ cell line replicates per condition). Numbers in the left plot indicated the number of enriched genes per functional term. Bold terms were considered to be related to autophagy. The righthand-side plot illustrated whether the genes causing the enrichment of the functional terms were overexpressed on both the transcriptional and protein levels (blue) or only on the protein level (orange). (B) Fold change of key autophagy regulators in RNA-seq contrasting URE vs WT LSKs ($n = 4$ mice per condition). (C) Western blot towards the autophagy marker LC3B. Boxed lanes (representative image) were used for quantification and as input for the barplot representation on the righthand side. VCP was used as a loading control. Significance was assessed by an unpaired $t$-test ($n = 3$, independent cell line replicates). (D) Assessment of autophagic flux by Cyto-ID flow cytometry. Significance was calculated by an unpaired $t$-test ($n = 5$, independent cell line replicates). A representative histogram was shown for Hox-WT and Hox-URE cell lines. Barplot indicated per-replicate normalized fluorescence intensity. Significance by an unpaired $t$-test ($n = 5$, independent cell line replicates). The horizontal lines represented the median, upper and lower hinges represented the 25th and 75th quartile, and whiskers represented data no more distant from the whiskers than 1.5 times the interquartile range. All data points were additionally shown as dots. (E) Assessment of autophagic flux by Cyto-ID in Hox-WT and Hox-URE cells after transduction with an anti-Vmp1 or control shRNA. The significance of greater autophagy reduction upon Vmp1 knockdown in Hox-URE compared to Hox-WT cells was assessed by two-way ANOVA ($n = 5$, independent cell line replicates). (F) Assessment of survival over time in Hox-WT and Hox-URE cells cultured in the presence of two different concentrations of the Bcl2 inhibitor Venetoclax or DMSO vehicle control. The significance of the greater reduction in survival over time in Hox-URE vs Hox-WT cells was assessed by two-way ANOVA and corrected for multiple testing by Benjamini–Hochberg ($n = 3$ independent cell line replicates per treatment along the time course). Error bars represent standard deviations. (G) Assessment of autophagic flux by Cyto-ID for Hox-URE cells cultured in the presence of Flt3 ligand after transduction with either a pMIG-Flt3 overexpression or empty vector control construct. Three independent biological replicates (transductions) were performed, each with 3–6 culture replicates based on the same transduction. Significance was assessed by one-way ANOVA based on the average of the culture replicates ($n = 3$). (H) Assessment of autophagic flux by Cyto-ID in Hox-URE cells transduced with an estrogen-inducible PU.1 overexpression construct or empty vector, followed by culture in an estrogen-containing medium. Significance was assessed by an unpaired $t$-test ($n = 3$). (E, G) The horizontal line represented the median, upper and lower hinges represented the 25th and 75th quartile, whiskers represent data no more distant from the whiskers than 1.5 times the interquartile range, outliers beyond the whiskery were not shown. All data points were additionally shown as dots. Source data are available online for this figure.

proteomics data for functional enrichment analysis of proteins overexpressed in Hox-URE as compared to Hox-WT cells (Dataset EV4). While the top enriched term was "Neutrophil degranulation", in line with the enhanced neutrophil character of URE$^\Delta$ cells, we indeed identified enrichment of protein pathways directly annotated in autophagy and mitophagy processes (Fig. 5A—left panel; Dataset EV4). For the term "Autophagy—animal", this included genes such as *Autophagy Related 7 (Atg7)*, *Lysosomal-Associated Membrane Protein 1* and *2 (Lamp1/2)* as well as *Protein Kinase CAMP-Activated Catalytic Subunit Alpha* and *Beta (Prkaca/b)* and MAP kinases *(Mapk3, Mapk9)*. The term "Mitophagy – animal" comprised genes with known functions in autophagy induction and regulation such as *Fission1 (Fis1)* (Waters et al, 2022), the mitophagy receptor *Bcl2l13* (Murakawa et al, 2015) and the TF *Foxo3* (Audesse et al, 2019). In addition, we found enrichments for terms whose gene products might contribute to autophagy, such as "Membrane Trafficking" (Søreng et al, 2018), "Endocytosis" (Tooze et al, 2014), "Lysosome" (Hu et al, 2015) as well as "Peroxisome" and "Peroxisomal protein import" (Li et al, 2021), hence terms related to the trapping, engulfment or degradation of (intracellular) components.

Bulk RNA-seq comparing ex vivo URE$^\Delta$ vs WT LSKs confirmed that key components of autophagy regulation were overexpressed in early URE$^\Delta$ cells also at the transcriptional level, including *Foxo3*, *Dapk1* and others (Fig. 5B). At the genome-wide level, however, about half of the genes included in autophagy-related terms were enhanced at both the transcript and protein levels, while the other half was enhanced exclusively as protein (Fig. 5A—right panel). These findings illustrate the importance of multi-omics approaches to comprehensively investigate molecular mechanisms, and they suggest that processes downstream of the transcriptional machinery might notably contribute to enhanced autophagy of PU.1-downregulated cells.

We confirmed stronger autophagy in Hox-URE$^\Delta$ than in Hox-WT cells by detecting higher levels of autophagosome-associated protein LC3-II (as measured by Western blotting) and an increased

autophagic flux as measured by Cyto-ID staining in flow cytometry, (Fig. 5C,D). Moreover, we transduced both Hox-WT and Hox-URE cells with an anti-Vmp1 shRNA and then measured Cyto-ID staining, confirming the notable dependency of URE$^\Delta$ cells on Vmp1 while effects in WT cells were modest (Fig. 5E).

Taken together, PU.1 downregulation causes comprehensive transcriptomic and proteomic changes that enhance cellular autophagy.

## PU.1-downregulated cells depend on Bcl2 upregulation

Excessive autophagy can induce cell death (Jung et al, 2020). However, Bcl2 has been shown to block autophagy-associated cell death (Marquez and Xu, 2012). Hence, URE$^\Delta$ cells may avoid this fate by upregulating Bcl2, which was part of signature 3 and was identified as a gene whose knockdown preferentially inhibited Hox-URE cell growth. To test this issue, we treated both Hox-URE and Hox-WT cells with the clinically used Bcl2 inhibitor Venetoclax. Of note, Hox-URE but not Hox-WT cells showed a notable dependency on Bcl2 towards survival (Fig. 5F), providing evidence that URE$^\Delta$ cells may suppress cell death to cope with enhanced autophagy.

## Restored FLT3 signaling reduces autophagy

URE$^\Delta$ cells failed to properly express several genes encoding cytokine receptor and signaling components (signatures 1 and 2 in Fig. 2A), including the gene encoding Flt3, which is an important receptor for HSPCs (Appendix Fig. S4B) (Gilliland and Griffin, 2002). Therefore, we reasoned that enhanced autophagy might serve as a failsafe mechanism to maintain the survival of PU.1-downregulated cells in spite of their reduced ability to respond to certain cytokines. To test this concept, we retrovirally re-expressed Flt3 in Hox-URE$^\Delta$ cells followed by Flt3 ligand (Flt3l) treatment. Indeed, restored Flt3 signaling impaired autophagy, supporting a potential link between autophagy and cytokine signaling (Fig. 5G).

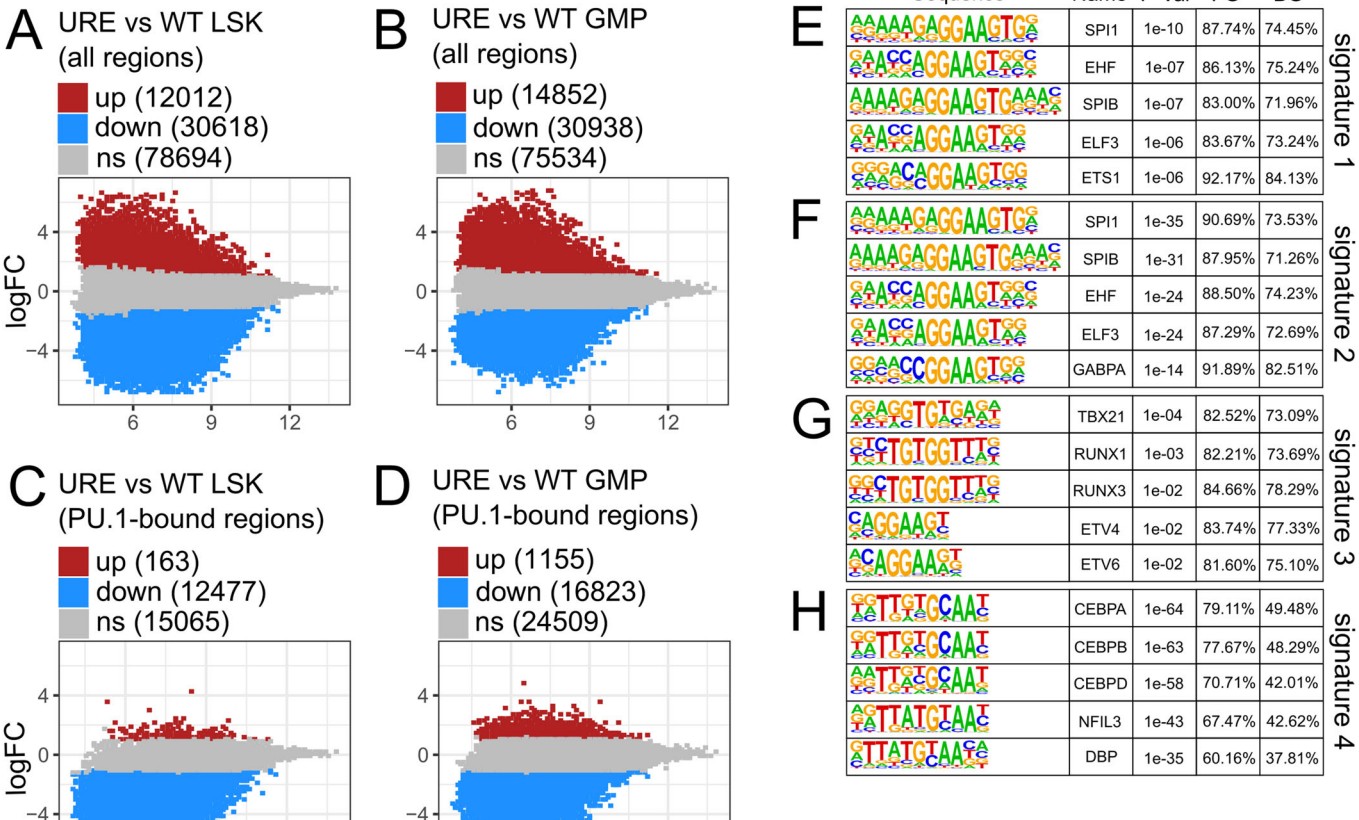

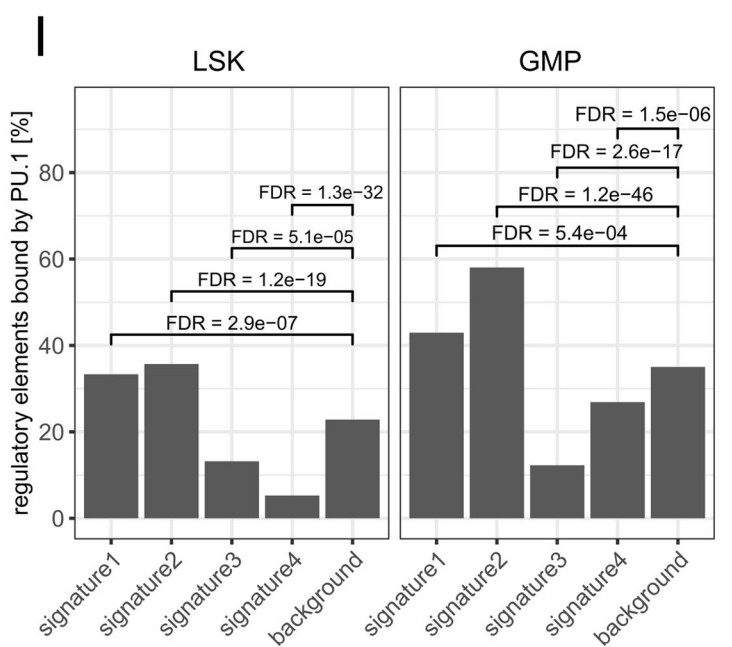

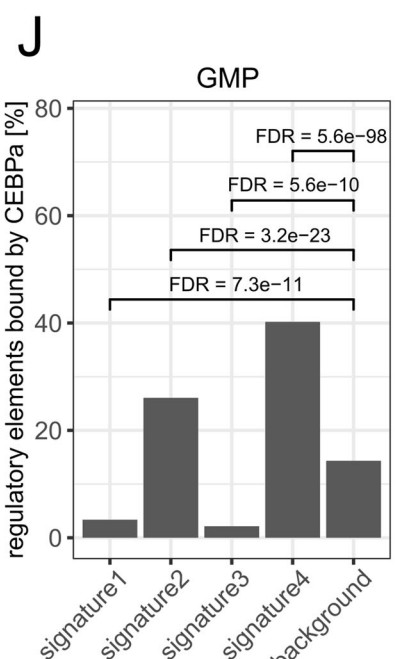

up: more accessable in URE
down: more accessable in WT

**Figure 6.  PU.1 represses genes in myeloid progenitors indirectly.**

(A–D) MA-plots summarizing differential accessibility analysis between URE and WT cells (LSK, GMP) by ATAC-seq (FDR < 0.05, fold change >2, $n = 3$ mice per condition). (A) URE vs WT LSKs considering all differentially accessible regions (DARs). (B) URE vs WT GMPs considering all DARs. (C) URE vs WT LSKs considering DARs bound by PU.1 in WT LSKs based on ChIP-seq (Pundhir et al, 2018), (D) URE vs WT GMPs considering DARs bound by PU.1 in WT GMPs based on ChIP-seq. (E–H) The top five most enriched motifs for DARs assigned to signature 1–4 genes. FG/BG: Percent of foreground ( = test set) or background regions harboring the motif. (I) Percentage and enrichment statistics for DARs assigned to genes from signatures 1–4 and bound by PU.1 in WT LSKs (left) or GMPs (right) based on ChIP-seq. (J) As in (I) but for CEBPa in WT GMPs based on ChIP-seq. Significances in (I, J) comparing percentages of TF_bound regions with the background were calculated with a Chi-square test and corrected for multiple testing by Benjamini–Hochberg.

Likewise, the re-expression of PU.1 in Hox-URE$^\Delta$ cells via an estrogen-inducible PU.1-ER construct significantly reduced autophagy, confirming the connection between low PU.1 levels and enhanced autophagy in URE$^\Delta$ cells (Fig. 5H).

## PU.1 downregulation overactivates genes indirectly

To determine the mechanism of how genes are overexpressed in URE$^\Delta$ cells, we produced genome-wide accessibility data using the assay for transposase-accessible chromatin (ATAC-seq) from sorted LSKs and GMPs. Accessible ( = open) chromatin regions identified by ATAC-seq are commonly interpreted as putative gene regulatory elements (REs) (Buenrostro et al, 2013). We mapped between 70,000 and 80,000 callable open chromatin regions in each of the four experimental groups (WT-LSK, URE$^\Delta$-LSK, WT-GMP, URE$^\Delta$-GMP, Dataset EV5), which we merged to obtain the final "regulome" of 121,336 genomic sites. We analysed this "regulome" for differential accessibility between WT and URE$^\Delta$ cells with DESeq2 (Love et al, 2014). We identified 42,630 differential regions between URE$^\Delta$ and WT LSKs and 45,790 differential regions between URE$^\Delta$ and WT GMPs, representing 35–37% of all open chromatin regions (Fig. 6A,B). In both comparisons, two to three-fold more regions showed decreased accessibility in the URE$^\Delta$ cells, in line with previous reports (Minderjahn et al, 2020; Watt et al, 2021) suggesting that on a genome-wide scale, PU.1 down-regulation leads to fewer active regulatory elements.

Next, we intersected the open chromatin regions with genome-wide PU.1 binding data from published chromatin immunoprecipitation followed by sequencing (ChIP-seq) of LSKs (40,085 bound sites) and GMPs (71,070 sites) (Pundhir et al, 2018). In WT LSKs, a total of 27,705 open chromatin regions were PU.1-bound, of which 12,477 (45%) showed decreased accessibility in URE$^\Delta$-LSKs while only 163 regions (0.5%) showed increased accessibility (Fig. 6C). Analogously, a total of 42,487 open chromatin regions were PU.1-bound in WT GMPs, of which 16,823 (40%) showed decreased accessibility in URE$^\Delta$-GMPs while 1155 (2.7%) showed increased accessibility (Fig. 6D). This highlights that in both LSKs and GMPs, PU.1 binding is almost exclusively associated with the maintenance of chromatin accessibility.

This mutual exclusion between PU.1 binding and the lack of chromatin closing of PU.1-bound sites upon PU.1 downregulation suggests that enhanced chromatin accessibility and gene overexpression in URE$^\Delta$ cells was indirect of PU.1. To address whether other transcription factors are involved, we first assigned putative REs to signature genes (see Methods) and scanned these REs for motif enrichment with HOMER (Heinz et al, 2010). Signatures 1 and 2 were enriched for Ets factor motifs, with the Spi1 (PU.1) motif being the top hit (Fig. 6E,F). Accordingly, ChIP-seq data

(Pundhir et al, 2018) confirmed enriched PU.1 occupancy at signature 1 and 2 REs in both WT LSKs and WT GMPs (Fig. 6I), suggesting that signature 1 and 2 genes (which were under-expressed in URE$^\Delta$ cells) are direct PU.1 targets.

Conversely, signature 3 and 4 REs were depleted for PU.1 binding, confirming that these signature genes are indirect PU.1 targets (Fig. 6I). Signature 4 was enriched for Cebp motifs and showed enriched Cebpa binding using published ChIP-seq data from WT-GMPs (Jakobsen et al, 2019) (Fig. 6H,J). Hence, signature 4 is likely downstream of Cebps, which is in line with the predominance of neutrophil genes shaping this signature. Finally, signature 3 was the most strongly PU.1 binding-depleted signature in GMPs, and was enriched for Tbx, Runx, and Ets (Etv4/6, Erg, but not PU.1) motifs (Fig. 6G,I).

Collectively, our data suggest that while gene activation in HSPCs is directly downstream of PU.1, gene repression is indirect and requires other TFs.

## PU.1 downregulation causes specific remodeling of RUNX1 chromatin binding in myeloid cells

RUNX1 and C/EBPα are known binding partners of PU.1 (Zhang et al, 1996; Imperato et al, 2015). Our ATAC data revealed a genome-wide shift from the PU.1 motif to enriched RUNX1 and CEBPα motifs within the open chromatin of both URE$^\Delta$-LSKs and URE$^\Delta$-GMPs (Fig. 7A), suggesting that PU.1 downregulation leads to globally redistributed binding patterns of its partner TFs. Because the RUNX motif was enriched at signature 3 genes, we generated myeloid PU.1 level-dependent genome-wide RUNX1 chromatin binding data. As we failed to produce high-quality results with commercially available antibodies against mouse RUNX1, we conducted RUNX1 ChIP-seq with the above-mentioned human THP-1 AML cells following induction of shRNAs against PU.1 or a scramble control. Indeed, PU.1 knockdown grossly redistributed RUNX1 chromatin binding (Appendix Fig. S5A,B). Overlaying RUNX1 with PU.1 ChIP-seq peaks (the latter derived from parental THP-1 cells (Minderjahn et al, 2020)), indicated that PU.1 knockdown shifted RUNX1 binding from previously PU.1-cobound genomic sites to regions lacking PU.1 binding in the parental cells (Fig. 7B). Notably, a similar PU.1-dependent remodeling of RUNX1 binding has recently been described in early T cells (Hosokawa et al, 2018). Hence, PU.1 downregulation in myeloid cells leads to lineage-inappropriate RUNX1 binding mimicking a pattern normally seen in thymocytes.

To better characterize the genomic loci with increased or decreased RUNX1 binding upon PU.1 knockdown, we performed motif enrichment analysis and integrated them with published

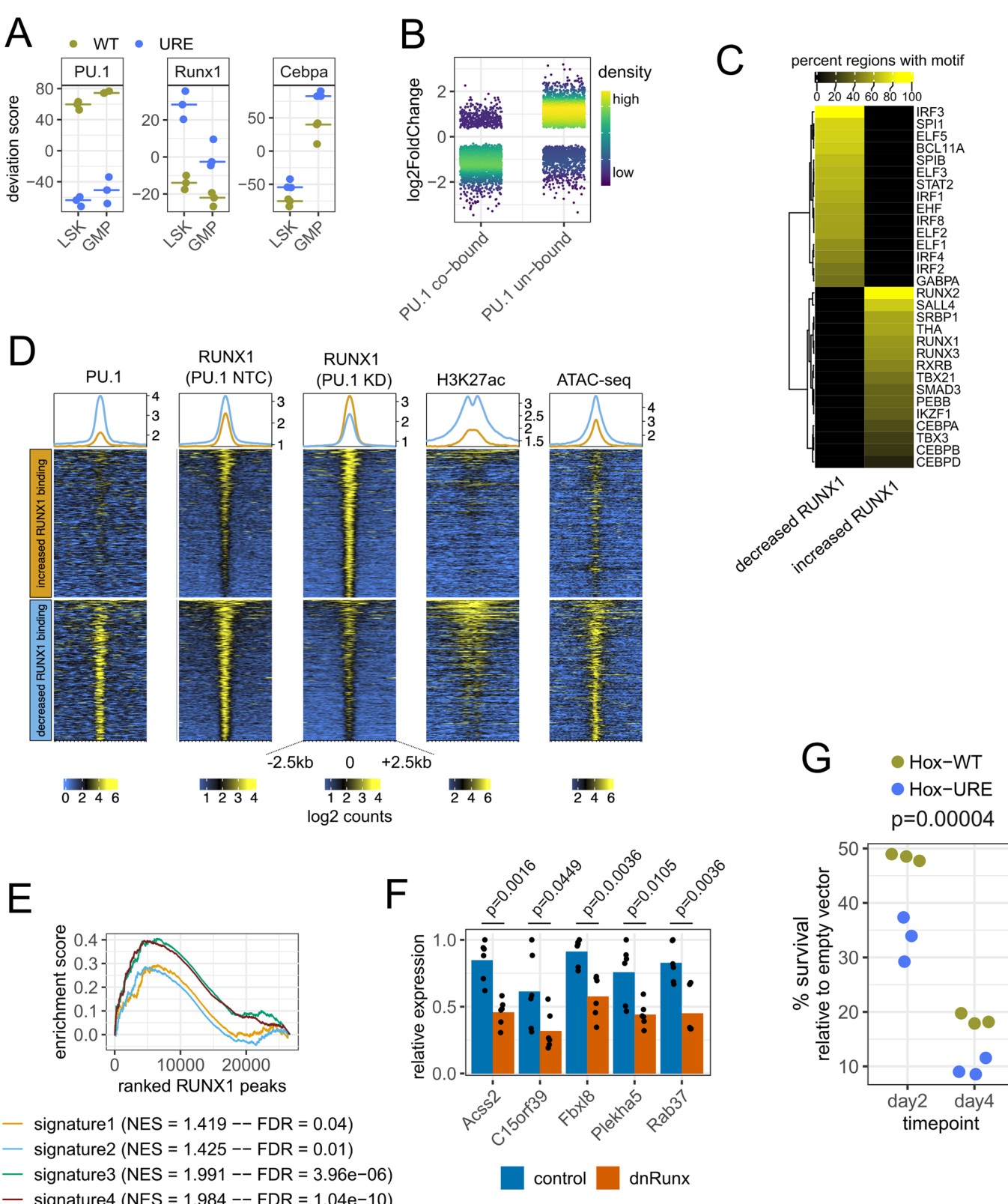

◀ **Figure 7. RUNX1-mediated gene expression regulation in PU.1-downregulated cells.**

(A) Regulome-wide motif accessibility deviation scores based on ATAC-seq data for the indicated motifs and cell types calculated by chromVAR ($n = 3$ mice per condition). Horizontal bars indicate the median (B) Fold changes of significantly different (FDR < 0.05, R/Bioconductor package DESeq2) RUNX1 binding between PU.1-knockdown and control THP-1 cells by ChIP-seq ($n = 6$ cell line replicates with independent immunoprecipitations and library preparations). Differential regions were split by co-binding events with PU.1 in parental THP-1 cells based on published ChIP-seq data (Minderjahn et al, 2020). (C) Motif enrichment analysis for loci with increased or decreased RUNX1 binding after PU.1 knockdown. The heatmap shows the percent of regions towards significantly enriched (FDR < 0.05) motifs for the two sets of genomic sites. (D) Genomic heatmaps for loci with increased (top) or decreased (bottom) RUNX1 binding after PU.1 knockdown in a 5 kb window centered at the RUNX1 peak center. Signal intensities represent log2 read counts towards (from left to right) PU.1 ChIP-seq (GSE128834), RUNX1 ChIP-seq from this study in THP-1 cells with NTC or anti-PU.1 shRNA transduction, H3K27ac ChIP-seq (GSE117864) or ATAC-seq (GSE96800). All published datasets were created from parental / untreated THP-1 cells. (E) Gene set-like analysis for differential RUNX1 binding in PU.1 knockdown and control THP-1 cells. RUNX1-bound regions cobound by PU.1 in parental THP-1 cells in a 50 kb window around the TSSs of signatures 1–4 from the scRNA-seq analysis were defined as "gene sets" and analysed by gene set enrichment analysis for global binding shifts. Permutation-based FDRs and normalized enrichment scores (NES) for each signature were calculated with the R/Bioconductor package fgsea. (F) Gene expression changes in URE-AML cells after transfection with either a dominant-negative Runx-family binding domain (dnRunx) or empty vector control. Significances were calculated by one-way ANOVA and corrected for multiple testing using Benjamini–Hochberg ($n = 6$). (G) Survival assay in Hox-WT and Hox-URE cells after transduction with a dnRunx construct. The significance of decreased survival in Hox-URE compared to Hox-WT across the two timepoints after transduction was assessed by one-way ANOVA and corrected for multiple testing using Benjamini–Hochberg ($n = 3$). Source data are available online for this figure.

PU.1 and H3K27ac ChIP-seq as well as ATAC-seq data from parental THP.1 cells (Phanstiel et al, 2017; Godfrey et al, 2019; Minderjahn et al, 2020). Loci with reduced RUNX1 occupancy after PU.1 knockdown showed preferences for known PU.1 binding partners such as IRF and ETS factors, whereas regions with increased RUNX1 occupancy showed preferences for RUNX, Spalt-like, Ikaros, and CEBP-family (Fig. 7C). In line with our analysis of signature 3 which was enriched for RUNX1 but depleted of PU.1 binding, those loci with increased RUNX1 occupancy after PU.1 knockdown were generally deprived of PU.1 binding (Fig. 7D). Interestingly, in normal THP-1 cells those loci globally showed lower levels of the activating histone mark H3K27ac and decreased chromatin accessibility compared to loci with reduced RUNX1 binding. This suggests a model in which under normal myeloid conditions these genes are not fully activated but might be poised for activation to ensure cellular survival upon emergency conditions (such as PU.1 downregulation).

To further explore this model, we tested whether enhanced RUNX1 binding to PU.1-unbound chromatin sites in the PU.1 knocked down human THP-1 cells was linked to genes overexpressed in URE$^\Delta$ mice. To this end, we filtered for all RUNX1 peaks within ±50 kb windows of the signature 1–4 transcription start sites (TSS) that were not cobound by PU.1, and used them to define signature-specific single RUNX1-bound "gene sets" for GSEA (Fig. 7E). This confirmed that when PU.1 was knocked down, RUNX1 globally bound more strongly to sites normally not cobound by PU.1. Importantly however, it revealed higher enrichment scores and lower false discovery rates (FDRs) for PU.1-repressed gene signatures $3 + 4$ than for the PU.1-activated signatures $1 + 2$ in the PU.1 knockdown cells, indicating that enhanced RUNX1 binding to PU.1-unbound regions is preferred at genes overactivated by PU.1 knockdown.

### PU.1-downregulated cells have a greater need for RUNX activity

We next employed published ChIP-seq data from primary human AML patient cells (Ptasinska et al, 2014) to assess whether evidence exists for RUNX1 binding to signature 3 genes for which we could demonstrate essential pro-survival functions in the URE-AML model. Even though we applied a stringent window of ±1 kb around the TSS to avoid false assignments in identifying human-to-mouse

distal RE homologs, we found that 66 (41%) of the 160 human genes (of the 178 mouse genes which had an annotated human homolog) showed RUNX1 binding. Notably, RUNX1 but not PU.1 bound to at least five out of the six PU.1-repressed genes overexpressed in the human BeatAML cohort (ACSS2, C15ORF39, FBXL8, RAB37, and VMP1; Appendix Fig. S5C).

Retroviral transduction of URE-AML cells with a dominant-negative form of RUNX (dnRUNX, (Sun et al, 2001)) to block activity of RUNX1 (and potentially redundant RUNX2/3), led to down-regulation of the aforementioned genes (Fig. 7F), providing proof that RUNX is required for their expression. Finally, overexpression of dnRUNX in Hox-WT and Hox-URE cells demonstrated that despite its expected general relevance for the survival also of WT cells, reduced RUNX activity had a greater effect on the survival of URE$^\Delta$ cells, thus providing evidence for the increased dependency of PU.1-down-regulated cells on RUNX function (Fig. 7G).

## Discussion

Decreased PU.1 levels have been shown to drive excessive HSC divisions (Staber et al, 2013). However, excessive cell divisions cause substantial cellular stress, which normally would lead to rapid HSC exhaustion. HSCs are protected against such stress-mediated exhaustion by their ability to induce a robust autophagy response, a feature that is lost upon differentiation into short-lived myeloid progeny (Warr et al, 2013a). Our data indicate that downregulated PU.1 fails to restrict pro-autophagy genes in HSCs and confers enhanced autophagy onto myeloid progenitors, suggesting that PU.1 normally terminates autophagy during myeloid differentiation. Because a positive correlation exists between autophagy and longevity (Nakamura and Yoshimori, 2018), the inability of PU.1-downregulated cells to terminate autophagy could result in an enhanced cellular life span and a corresponding higher likelihood of accumulating mutational burden over time, resulting in clonal expansion of fully malignant progeny. Hence, our data suggest the yet undescribed possibility that autophagy represents a pro-oncogenic process in non-transformed cells.

Cytokine starvation is a potent inducer of HSC autophagy (Warr et al, 2013b). Although HSPCs from adult PU.1 knockout or URE$^\Delta$ mice are still able to grow in response to IL-3, SCF, and G-CSF, they are unable to respond to M- and GM-CSF signaling (DeKoter

et al, 1998; Rosenbauer et al, 2004; Nutt et al, 2005). Our single-cell transcriptomics data support the view that PU.1 downregulation leads to grossly impaired cytokine pathways, because URE$^\Delta$ cells underexpressed many cytokine receptors (including components of the receptors for Il-1, Il-4, Il-6, and Il-10 as well as TNF family including the TNF ligand APRIL (Tnfsf13), its receptor Tnfrsf13b and the TNFα receptor Tnfrsf1a, and also Flt3). Hence, a limited capacity to respond to the full spectrum of cytokines and growth factors is likely one reason why URE$^\Delta$ cells upregulate a large autophagy-predominated gene signature, for the majority of genes of which we could functionally demonstrate essential roles for cell survival. This link between reduced cytokine response and enhanced autophagy is supported by our Flt3 reconstitution experiment, which reduced autophagy.

While excessive autophagy can induce cell death (Jung et al, 2020), our data indicate that URE$^\Delta$ cells actively avoid this fate by upregulating Bcl2. Bcl2 balances beneficial and detrimental impacts of autophagy on cell survival by binding the autophagy protein Beclin 1 (Marquez and Xu, 2012). Hence, Bcl2 limits an overshoot autophagy activation and, thus, inhibits autophagy-associated cell death (Xu et al, 2013).

Data from the URE$^\Delta$ model suggest that enhanced autophagy in AML cells can be inherited from the preleukemic cells of origin. In a clinical context, this might imply that screening for increased autophagy could identify subpopulations of non-transformed "healthy" cells that are prone or "at risk" of developing into a malignant state. Along the same line, it is remarkable that the PU.1-repressed autophagy genes that we found upregulated in human AML cells were those that in our shRNA screen were identified as essential for the survival of both preleukemic and leukemic URE$^\Delta$ cells, suggesting that they are important throughout leukemogenesis. Consequently, one could conclude that at least some AML subtypes appear to still depend on those altered survival programs that are manifested early during disease development. Our finding that these genes are essential for the survival of URE$^\Delta$ AML but not healthy control cells indicates that they might represent interesting targets to preferentially disrupt the survival of diseased over healthy cells.

Overactivation of the survival gene signature 3 and the neutrophil signature 4 occurred concurrently within single URE$^\Delta$ HSPCs. This suggests that both signatures might be linked by a common overarching regulatory mechanism. Indeed, both transcriptional programs are controlled by an indirect PU.1 repressor function. Overactivation of signature 4 correlated with enhanced CEBPα binding, a well-established master regulator of neutrophil development (Nerlov, 2007). Interestingly, while signature 4 REs showed enriched CEBPα binding, they showed depletion of PU.1 binding. This suggests that CEBPα binds signature 4 REs without PU.1. In contrast, REs of the monocytic signature 2 (which was underexpressed in URE$^\Delta$ cells) demonstrated enriched binding of both CEBPα and PU.1, indicating that these genes are driven by a combined action of CEBPα with PU.1. Congruently, it appears that in the absence of sufficient PU.1 levels in URE$^\Delta$ cells, CEBPα binding shifted from composite PU.1-CEBP sites to sites which (in WT cells) are not bound by PU.1. Consequently, CEBPα changed from controlling monocytic genes (together with PU.1) to activating neutrophil genes (without PU.1).

Interestingly, a similar scenario has recently been shown to occur in T-cell development. Rothenberg and colleagues have found that PU.1 "steals" partner TFs (in particular RUNX1 and SATB1) from sites at which they would bind alone (without PU.1) and tethers them to composite binding sites to which PU.1 also binds (Hosokawa et al, 2018). As PU.1 is downregulated during normal thymocyte development, the above-described regulatory model implies that PU.1 partner TFs are freed to shift from composite sites to genomic sites to which they can bind alone, thus inducing T cell-specific genes. Our results now extend this mechanism to monocytic versus neutrophil lineage choice in HSCs, and to the context of leukemogenesis. In particular, we present evidence that PU.1 level-dependent remodeling of RUNX1 from RUNX/PU.1 composite binding sites towards RUNX single sites could constitute a pro-oncogenic mechanism, linking a PU.1-dependent sift in RUNX1 binding to (pre)leukemic cell growth via upregulation of pro-autophagy genes. This model fits well with the ascribed pro-survival role of RUNX1 in myeloid leukemia (Goyama et al, 2013).

Collectively, we link reorganized TF chromatin binding patterns as caused by the loss of a partner TF (in our case PU.1), to reprogrammed differentiation and cellular survival in leukemogenesis. Hence, when TFs are considered as therapeutic targets in disease treatments, such interoperable actions should be taken into consideration.

# Methods

**Reagents and tools table**

| Reagent/resource | Reference or source | Identifier or catalog number |
|---|---|---|
| **Experimental models** | | |
| C57BL/6 J URE$^\Delta$ mice | The Jackson Laboratory | 006083 |
| Hox-WT cells | Fischer et al 2019 (https://doi.org/10.1038/s41590-019-0343-z) | N/A |
| Hox-URE, URE-AML cells | established in-house, the first publication in this study | N/A |
| THP-1 parental cells | ATCC | TIB-202 |
| THP-1 cells with PU.1 inducible knockdown | Schuetzmann et al, 2018 (https://doi.org/10.1182/blood-2018-02-834721) | N/A |
| **Recombinant DNA** | | |
| pRSI17-U6-sh-UbiC-TagGFP2-2A-Puro shRNA expression plasmid | Cellecta | SVSHU617-L |
| pMIG cDNA expression plasmid | Holmes et al 2006 (https://doi.org/10.1101/gad.1396206) | N/A |
| **Antibodies** | | |
| anti-PU.1 | Santa Cruz | sc-390405X |
| anti-VCP | abcam | ab11433 |
| anti-RUNX1 | abcam | ab272456 |
| IgG isotype control (for anti-RUNX1) | Cell Signaling | 6636S |

| Reagent/resource | Reference or source | Identifier or catalog number |
|---|---|---|
| **Oligonucleotides and other sequence-based reagents** | | |
| shRNA screen cloning and sequencing oligos | Please see Dataset EV3 for a full sequence documentation. | |
| qPCR primers (murine/human) | Please see Dataset EV6 for a full sequence documentation. | |
| **Software** | | |
| R | CRAN | 4.0.3 |
| DESeq2 | Bioconductor | 1.30.0 |
| limma | Bioconductor | 3.46.0 |
| MAGeCK | SourceForge | 0.5.9.5 |
| **Other** | | |
| Illumina Nextseq 500 | | |
| Illumina Nextseq 2000 | | |
| Illumina HiSeq 3000/4000 | | |
| Illumina Novaseq 6000 | | |

## Mice and FACS sorting of hematopoietic progenitors

URE$^\Delta$ mice were described previously (Rosenbauer et al, 2004) and sacrificed (littermates, C57BL/6 background) at 8–12 weeks of age using $CO_2$ inhalation. Femurs were flushed and then lysed with ACK. c-Kit$^+$ cells were pre-enriched using Cd117 microbeads (Miltenyi) and stained with 7-AAD, lineage (Cd45r, Cd3, Cd4, Cd8, Cd19, Ter119, and Gr1), c-Kit, Sca-1, and CD16/32 (flow antibodies see Appendix Methods). We sorted Lin$^-$/c-Kit$^+$ cells for scRNA-seq, Lin$^-$Sca-1$^+$c-Kit$^+$ (LSKs), and Lin$^-$Sca-1$^-$c-Kit$^+$Cd16/32$^+$ (GMPs) for RNA-seq and ATAC-seq. Animal experiments were approved by the German Federal Animal Protection Act (regulation number 81-02.04.2019.A187).

## Cell lines and culture

Cell lines were maintained at 37 °C / 5% $CO_2$. For Hox-WT, Hox-URE, and URE-AML cells, we used IMDM (Thermo), 20% FCS Superior (Biochrom), 100 µg/ml penicillin/streptomycin (P/S) (Roth), 2 µM ß-estradiol (Sigma), and 10 ng/ml of SCF, IL-3, IL-6 (Peprotech). The Hox-immortalized cell lines were created from c-Kit$^+$ isolated hematopoietic progenitors of a WT and URE mouse as described previously (Wang et al, 2006). URE-AML cells were established from AML blasts of a URE$^\Delta$ mouse by rounds of culture in IL-3-containing semi-solid and liquid medium until cells survived in isolation indefinitely. For THP-1 cells with inducible (doxocycline, dox) PU.1 knockdown or scramble controls (shRNA) (Schuetzmann et al, 2018), we used RPMI, 10% FCS, 100 µg/ml P/S, and 1% L-glutamine. Dox (20 ng/ml) treatment was performed for 72 h before running a downstream experiment.

## Autophagic flux measurement

Autophagic flux was measured using the Cyto-ID Autophagy Detection kit (Enzo Life Sciences) via FACS. For measurement upon Flt3 overexpression the *Flt3* coding sequence was cloned into pMIG (gift from S. Nutt) and transduced into Hox-URE cells via

retrovirus. After 5 days of recovery cells were treated with 100 ng/ml Flt3 ligand for 4 h followed by Cyto-ID measurement. Significances were calculated by one-way ANOVA with the experimental batch as additional covariate.

## Western blot

Western blots were stained for murine PU.1 (Santa Cruz sc-390405X, 1:1000), LC3B (Cell Signaling 3868S, 1:1000), or VCP (abcam ab11433, 1:5000).

## RT-qPCR

cDNA was quantified (Power SYBR Green Mix, Thermo 4368577) house-keeping controls (primers see Dataset EV6). Significances were calculated with one-way ANOVA on the ddCt values, adding the experimental batch (if present) as a covariate.

## Individual shRNA-mediated knockdown

URE-AML cells were transduced via lentivirus with shRNAs cloned into pRSI17-U6-sh-UbiC-TagGFP2-2A-Puro (Cellecta). Every gene was independently targeted by two unique shRNAs. Sequences of qPCR primers and shRNAs are available in Datasets EV3 and EV6. We used these shRNAs:

ENSMUSG00000027605.18_Acss2__[1, 4], ENSMUSG00000033313.11_Fbxl8__[2, 3],

ENSMUSG00000018171.9_Vmp1__[2, 5], Rpl30__[1, 2] (killing control), NTC__[11, 16] (NTC)

## dnRunx overexpression

A HA-tagged dominant-negative Runx-family binding domain ((Sun et al, 2001), DNA sequence see Appendix Material & Methods) (or empty vector control) was cloned into pMIG and transduced via retrovirus into target cells. After recovery, RNA was extracted, and qPCR was performed (Primers in Dataset EV6).

## scRNA-seq

Lin$^-$/c-Kit$^+$ from WT and URE mice ($n = 2$ mice per genotype) were subjected to scRNA-seq using the 10x Chromium v3 gene expression kit following the manufacturer's instructions. Ten thousand cells were loaded into four reactions, respectively (one reaction per mouse). Libraries were sequenced to >50,000 reads per cell. Raw counts were obtained with salmon-alevin (Srivastava et al, 2019) and normalized by deconvolution (Lun et al, 2016). Samples were integrated (fastMNN), clustered, low-quality cells, doublets, and dendritic cells (Cd74$^{hi}$) were removed. Differential expression was carried out on pseudobulk level with DESeq2. Pseudotime ordering was performed with slingshot for clusters C1, C2, and C3. Differential genes from URE vs WT comparisons were clustered by hclust. A detailed description of computational methods is provided in the Appendix Material & Methods.

## Bulk RNA-seq

Low-input RNA-seq was performed on LSKs and GMPs from WT and URE mice ($n = 4$ mice per genotype and celltype). Cells were

sorted into cold PBS with 2% FCS, and RNA was immediately isolated (Macherey Nagel RNA XS Plus kit). Either 10 ng (if available) or all RNA was used to construct unstranded libraries with the NEBNext Single-Cell/Low-Input RNA library prep. Stranded RNA-seq was performed on Hox-WT, Hox-URE, and URE-AML cell lines ($n = 3$ per cell line) with the NEBNext Ultra II RNA Directional library prep kit, using 1 µg of RNA for mRNA enrichment. Reads were quantified with salmon (Patro et al, 2017) and aggregated to gene level with tximport. Differential expression (FDR < 0.05, absolute fold change >1.5) was carried out with DESeq2. Gene set enrichment analysis was performed with the fgsea package. Differential expression in the BeatAML cohort was tested via limma-trend (Ritchie et al, 2015).

## ATAC-seq

ATAC-seq was performed on LSKs and GMPs from WT and URE mice ($n = 3$ mice per genotype and celltype). We followed the OmniATAC protocol (Corces et al, 2017), starting with 7000–35,000 cells depending on availability, with the exception that KAPA HiFi HotStart ReadyMix was used as polymerase. Reads were aligned to GRCm38 with bowtie2, duplicates and non-primary chromosomes were removed. Peaks per group (celltype + genotype) were called with Genrich (v0.6.1), and then DESeq2 was used for differential accessibility (FDR < 0.05, abs. fold change >2). Differential peaks were assigned to signature 1–4 genes by proximity and differential pattern criteria (see Appendix Material & Methods). Assigned regions were scanned for motif enrichment with Homer findPeaks.pl against a background set representing ATAC-seq peaks that, in all comparisons, consistently showed evidence against differential accessibility.

## ChIP-seq

ChIP-seq toward RUNX1 or IgG control in THP-1 cells with inducible PU.1 knockdown or scramble control was performed as described (Blecher-Gonen et al, 2013). Briefly, knockdown was induced, and cells were harvested after 72 h. A total of $2 \times 10^7$ cells were crosslinked in 1% formaldehyde, lysed, and then sonicated on a Bioruptor-200 at 4 °C. Ten micrograms of antibody (RUNX1 abcam ab272456 or IgG Cell Signaling 66362S) were used for IP bound to protein G magnetic beads. Libraries for Illumina were prepared with the NEBNext Ultra II DNA Library Prep kit. Two experimental rounds (batches) were performed. Reads were aligned to GRCm38 with bowtie2, duplicates, and non-primary chromosomes were removed. Peaks per condition and batch were called with macs2. Reproducible peaks between batches were identified with IDR, and then peaks from the two conditions (RUNX1 upon or without PU.1 knockdown) were merged and used as a template for a count matrix with featureCounts. Differential binding was assessed with DESeq2 (FDR < 0.05, abs. fold change >1.1). All overlap analysis was performed with the GenomicRanges package.

## Proteomics

The proteome of Hox-WT and Hox-URE cells was analyzed on a nanoflow HPLC (EASY-nLC1000, Thermo Fisher) coupled online to a Q Exactive HF-X Hybrid Quadrupole-Orbitrap Mass Spectrometer (Thermo Fisher). About 500 ng of peptides were used for analysis.

Details are provided in the Appendix Material & Methods. For analysis, only peptides were kept that had no more than two missing values in both genotypes. The remaining missing data were imputed with MSnSet (MinProb method). Differential analysis was carried out with limma (FDR < 0.05, abs. fold change >1.5). Overexpressed proteins were tested for enrichment (KEGG, REACTOME) with gost from gprofiler2.

## shRNA screen and validation

A detailed description of the shRNA screen procedures, target infection MOIs, library complexity/coverage, and analysis is provided in the Appendix Material & Method. Briefly, a focused lentiviral shRNA dropout screen against signature 3 genes from the scRNA-seq analysis was performed in triplicates in Hox-WT, Hox-URE and URE-AML cells. shRNA-attached barcodes were used to identify shRNAs in genomic DNA (gDNA) after viral delivery (screening vector pRSI17- U6-sh-UbiC-TagGFP2-2A-Puro from Cellecta). We included 344 of 346 signature 3 genes, as these allowed us to design five unique shRNAs per gene. A total of 30 established killing control shRNAs that were suggested by the company to work most efficiently in retarding cell growth (targeting the growth-essential genes Psma1, Rpl30, Polr2b with ten barcodes each), and 45 non-targeting (luciferase) controls were added. After lentiviral delivery, the gDNA of 50% of cells was harvested as input control after 24 h. The remaining cells were selected with puromycin and maintained until 10 cell divisions had happened, followed by gDNA harvest. Illumina sequencing libraries toward the barcodes were created by a two-step nested PCR and sequenced with 10% PhiX. Raw barcode counts were normalized to the NTC controls and analysed with MAGeCK RRA (v0.5.9.5) to call essential genes (negative permutation FDR < 0.01 in at least 3/5 shRNAs per gene). Log2 fold changes per shRNA against input libraries were calculated with limma-voom followed by quantile normalization (qn). Differential log2 fold changes of essential genes were obtained by subtraction of qn-normalized log2 fold changes between cell lines and thresholds using an empirical cutoff based on the NTCs. All shRNA, barcode, and primer sequences are provided in Dataset EV3 and EV6.

For the low-throughput validation we delivered individually cloned (same vector as in the screen) shRNAs to the target cells. GFP⁺ and GFP⁻ cells were sorted after 24 h and mixed at a 50/50 ratio. This ratio was monitored by FACS for indicated days. A decrease in GFP⁺ cells was interpreted as a knockout, causing reduced proliferation relative to the non-successfully transduced cells. NTCs and killing controls were used to define baselines.

## Sample size determination, binding, and randomization

No statistical methods were used to predetermine sample sizes. Commonly used sample sizes in the field were used. Experiments were not blinded or randomized. In case experiments required multiple experimental days, we ensured that replicates of each condition were included on every experimental day to avoid nested batch effects.

# Data availability

NGS data produced for this study are available at the Gene Expression Omnibus (GEO) under accession number GSE251674

Proteome data (MaxQuant output) are included in Dataset EV7. We used public ChIP-seq data towards PU.1 in murine LSKs and GMPs (GSE89767), CEBPA in murine GMPs (GSE118963), PU.1 in THP-1 (GSE128834), ATAC-seq in THP-1 (GSE96800), H3K27ac in THP-1 (GSE117864), and RUNX1 in AML blasts (GSE60130).

The source data of this paper are collected in the following database record: biostudies:S-SCDT-10_1038-S44318-024-00295-y.

## Code availability

Code for preprocessing and downstream analysis of NGS and proteome data will be made available upon publication at our GitHub repository (https://github.com/ATpoint/bender_et_al_2024). The code documentation contains all relevant software and package versions for the individual analysis steps.

## Peer review information

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

## Acknowledgements

We would like to thank Celeste Brennecka for their linguistic support. This work was supported by the Interdisciplinary Center of Clinical Research (IZKF grant Ros4-003-23) to FR and by the German Research Foundation (CRCTRR332 B03 and C07) to FR and TV. We thank the Core Genomics Facility Münster and the Biomedical Sequencing Facility Vienna for performing sequencing services.

## Author contributions

**Alexander Bender**: Conceptualization; Data curation; Formal analysis; Validation; Investigation; Visualization; Methodology; Writing—original draft; Project administration; Writing—review and editing. **Füsun Boydere**: Formal analysis; Validation; Investigation; Visualization; Methodology. **Ashok, Kumar Jayavelu**: Resources; Investigation. **Alessia Tibello**: Investigation. **Thorsten König**: Investigation. **Hanna Aleth**: Investigation. **Gerd Meyer zu Hörste**: Resources. **Thomas Vogl**: Resources. **Frank Rosenbauer**: Conceptualization; Resources; Formal analysis; Supervision; Funding acquisition; Methodology; Writing—original draft; Project administration; Writing—review and editing.

Source data underlying figure panels in this paper may have individual authorship assigned. Where available, figure panel/source data authorship is listed in the following database record: biostudies:S-SCDT-10_1038-S44318-024-00295-y.

## Funding

## Disclosure and competing interests statement

The authors declare no competing interests.

