## [Peer Review File · The EMBO Journal]

Redistribution of PU.1 partner transcription factor RUNX1 binding secures cell survival during leukemogenesis

Frank Rosenbauer, Alexander Bender, Füsün Boydere, Ashok Jayavelu, Alessia Tibello, Thorsten König, Hanna Aleth, Gerd Meyer zu Horste, and Thomas Vogl

Corresponding author(s): Frank Rosenbauer (frank.rosenbauer@ukmuenster.de)

Review Timeline:

Submission Date:	23rd Jan 24
Editorial Decision:	15th Apr 24
Revision Received:	29th Aug 24
Editorial Decision:	16th Oct 24
Revision Received:	21st Oct 24
Accepted:	23rd Oct 24

Editor: Cornelius Schneider

Transaction Report:

Dear Dr. Rosenbauer,

Thank you for submitting your manuscript for consideration by the EMBO Journal. We have now received comments from three reviewers, which are included below for your information.

As can be seen from the reports, all three referees found the results of importance and interest, agreed that the experiments were performed competently. However, there are also several major concerns which would need to be addressed before publication at The EMBO Journal. In particular we agree with both referees #2 and referee #3 that the novel link between PU.1 and autophagy is interesting but still descriptive and would benefit from additional experimental validation. I should also add that it is The EMBO Journal policy to allow only a single major round of revision and that it is therefore important to resolve the main concerns at this stage.

I am happy to answer any additional questions or discuss possible revisions via e-mail or videoconferencing.

We generally allow three months as standard revision time, which can be extended to 6 months in case of major revisions, such as the experiments required here. As a matter of policy, competing manuscripts published during this period will not negatively impact on our assessment of the conceptual advance presented by your study. However, we request that you contact the editor as soon as possible upon publication of any related work, to discuss how to proceed. Should you foresee a problem in meeting the deadline, please let us know in advance and we may be able to grant an extension.

Thank you for the opportunity to consider your work for publication. I look forward to your revision.

Yours sincerely,

Cornelius Schneider

Cornelius Schneider, PhD
Editor
The EMBO Journal
c.schneider@embojournal.org

We realize that it is difficult to revise to a specific deadline. In the interest of protecting the conceptual advance provided by the work, we recommend a revision within 3 months (14th Jul 2024). Please discuss the revision progress ahead of this time with the editor if you require more time to complete the revisions. Use the link below to submit your revision:

Referee #1:

In this manuscript Bender et al., describe a novel mechanism by which mouse myeloid progenitors survive in the presence of reduced levels of the lymphoid master regulator transcription factor PU.1 by upregulating pro-survival and autophagy related genes. The authors use a multi-omics approach combining single cell RNA-seq and proteomics to identify, among others, an upregulated gene signature in progenitor cells enriched for pro-survival genes conferring longevity to cells experiencing reduced levels of PU.1. They screened for the essentiality of these genes, and the most significant one belonged to the autophagy pathway. Knockdown caused reduced growth. The authors validated the upregulated expression of autophagy related genes in human Acute Myeloid Leukemia samples and found that the sustained activation of the autophagy pathway which is normally terminated in myeloid progenitors is crucial for the myeloid stem cells with low PU.1 to survive. The authors reasonably speculate that the sustained but not excessive activation of autophagy, increases longevity and thus the possibility to accumulate mutations, implying that autophagy could represent a novel pro-oncogenic process in non-transformed cells. ChIP-seq experiments demonstrated that the effect of PU.1 is indirect and more specifically that its reduced levels cause a redistribution of its partner TF RUNX1. Overall, the experiments are well thought and convincing, many of the findings are quite novel and the paper is well-written, although the Introduction needs to be expanded a little bit to better introduce the reader into the topic. Below are few suggestions to help the authors prepare a revised version of their manuscript.

Suggestions:

1. In figure 1B the authors should include a heatmap depicting the expression levels of marker genes in individual cells in addition to the provided averaged heatmap, so that readers can better appreciate the expression levels and the percentage of cells in which each marker is expressed. The same should be done for Figure 2A.
2. In Figure 1C the authors could add a dotplot depicting the expression of PU.1 in each individual cell as arranged in the UMAP, in addition to the already provided boxplots.
3. In figure 6(E-H) the percentage of regions in which each motif is found can be added, together with the respective abundance in background regions, so that readers can appreciate its enrichment in tested regions.
4. We believe that the available chip-seq and epigenomics data have not been analyzed and visualized in a combinatorial fashion to reach their full potential. Through, the use of heatmaps the authors could better illustrate the redistribution of RUNX1 peaks and what epigenomic markers and other TFs from the literature are co-enriched in these differentially bound sites in order to get a stronger sense of their regulatory potential.
5. The authors should add a detailed table containing quality metrics for ATAC-seq and ChIP-seq experiments.

Referee #2:

The major question investigated in this study is how preleukemic cells deficient in PU.1 are able to survive despite the critical

role of PU.1 in regulating HSC and lineage specific cell fate decisions. PU.1 loss was studied in the context of preleukemic HSPCs isolated from a previously published mouse model lacking a key Spi1 cis-enhancer termed upstream regulatory element (URE) in which the mice harbor downregulated Pu.1 expression and eventually develop AML. Overall, this is well performed study that is clearly written and structured. The main novelty of the work comes from the comparative transcriptomic, proteomic and epigenomic approaches to identify that PU.1-downregulated cells switch from cytokine-driven pathways to an autophagy gene signature consistent across the preleukemic and leukemic mouse models and existing AML patient datasets. This switch is mediated by the loss of PU.1 binding and the redistribution of the PU.1 binding partner, RUNX1. Major criticisms pertain to the very unclear methods descriptions and explanations of the shRNA screen analysis and results, lack of robust screen validation with individual shRNA, and overall descriptive nature of all presented "omics" data with no functional mechanistic proof. Because of these weaknesses, it is challenging to see how the shRNA screen informs PU.1 biology and unclear whether loss of PU.1 really creates a unique requirement for RUNX1 to induce autophagy to promote survival of URE cells. This work highlights a novel pro-oncogenic mechanism and reveals potential biomarkers that could be used to screen preleukemic cells at risk for leukemic transformation.

1. The conclusion from Figure 2 that "self-renewal and differentiation were interconnected" on the basis that the URE cells had decreased stem cell and myeloid genes by scRNA-seq analyses, is not fully supported. Would, for example, rescue of stem cell genes in the URE cells result in the upregulation of the same myeloid genes? Overexpression and/or knockdown experiments would greater support this claim. Why is C5 included in Figure 2A? It is not discussed.
2. Is Bcl2 the only anti-apoptotic gene that was upregulated in URE compared to WT in signature 3 in Figure 2A? It is postulated that the upregulation of Bcl2 seen in signature 3 and identified as an "essential" gene in the shRNA screen in URE cells may help these cells avoid apoptosis that may result from excessive autophagy. It would be beneficial to test whether inhibiting Bcl2 either with genetic approaches or pharmacological inhibitors (like Venetoclax) causes a reduction in autophagy to further support this potential role of Bcl2 balancing the "beneficial and detrimental impacts of autophagy on cell survival".
3. The shRNA screen, rationale, analysis plan, and data interpretation are incomplete or unclear. How was 10 cell divisions determined for each cell line? Why were only 3 positive control genes selected for and what is the evidence they are essential genes? Results text indicates shRNA pool was transfected. This is inaccurate, please revise. What was the MOI/infection efficiency of the pooled lentiviral library in each of the cell lines and what was the library coverage in the screen for each timepoint?
4. Results explain this screen readout as "knockdown" effect which is incorrect. Barcodes are measured and their abundance in the population corresponds to presumed effects of the attached shRNA on the cell its in given its change in abundance compared to input. Revise.
5. The description of the shRNA analyses is very confusing in both the results and methods text. The phrasing "more essential" is confusing (Fig. 3C) and how these genes are different than the essential genes in Fig. 3B that are unique to AML and URE is not clear. In Fig. 3B, there are 28 depleted shRNA unique to AML, 18 depleted shRNA unique to URE, and 18 depleted shRNA in common between AML and URE - and these are not overlapping with WT. Why then is the added step of differential fold changes required? In this analysis, were there no uniquely essential genes per any of the three cell lines? It is also confusing that "essential" is inconsistently used in quotations.
6. What is the scientific justification for terming a gene "more essential" in AML or URE cells versus WT? What is the functional and/or biological proof that a gene is more essential to one cell type versus another? How is this cut-off determined and measured for biological significance?
7. Fbxl8, Vmp1, and Acss2 were identified in the focused shRNA screen to be more essential for both URE and URE-AML cells compared to WT. In Figure 4D, of the 3 genes, Vmp1, is the only gene that visually appears to have an effect on survival similar to the killing control. Additionally, the percent survival graph of Acss2 doesn't appear very different compared to NTC. Was there any statistical testing performed in Figure 3D? If so, it would be valuable to include this to demonstrate a significant decrease in cell survival with each shRNA.
8. One of the major claims from the study suggests that autophagy is a pro-oncogenic process in non-transformed cells. Would inhibiting or impairing any of the autophagy factors identified in this work, such as Vmp1, prevent leukemic transformation in the Pu.1 URE mouse model? Does inhibiting Vmp1 alter autophagy in URE-specific cell lines? At present the relationship between PU.1 and autophagy descriptive.
9. The sentence "FBXL8, VMP1 and ACSS2, which were identified as "essential" in both Hox-URE and URE-AML but not Hox-WT cells in our shRNA screen" contradicts the finding from Figure 3 that claims that these three genes were considered "more essential for both Hox-URE and URE-AML than Hox-WT", implying that these genes are still essential in the Hox-WT cells. What was the need for calculating differential fold changes between AML and URE-Hox vs WT?
10. A more robust validation of more essential genes in AML and URE-Hox would be to use independent shRNA in all 3 cell lines, validate shRNA target repression in all 3 cell lines, and demonstrate that WT-Hox have unaffected growth by knockdown of FBXL8, VMP1, and ACSS2 as compared to URE-Hox and AML which are expected to have growth retardation.
11. The claim that human AML "preferentially overexpress PU.1-repressed genes that were required for both preleukemic and leukemic phases in the URE Δ model" is not convincing for all of the genes shown in Figure 4A. For example, for FBXL8, 10/13 AML subtypes have a DEG of 0.
12. It is unclear in Figure 4C what the gene expression changes are relative to in all of the gene expression plots, except for PU.1. The claim that PU.1 knockdown lead to reduced expression in the genes (pg.10-11) listed in Figure 4C is also not supported because the expression of ACSS2, C15orf39, FBXL8, PLEKHA5, RAB37, and VMP1 are all higher in the PU.1 KD compared to NTC.
13. The western blot in Figure 5C does not have the lanes labeled. It also unclear which bands were used for the normalized

intensity since there are two LC3B bands. Is VCP used as a loading control?

14. Flt3 was re-expressed in Hox-URE cells followed by stimulation with Flt3 ligand to impair autophagy. Is this a novel finding or are there any published studies that demonstrate Flt3 signaling having a role in relating autophagy? Suggesting a link between cytokine signaling and autophagy based on these data in Fig. 5E is premature.

15. Figure 6A-D should have the comparisons labeled on each plot as well as LSK and GMP. Are there overlap in ATAC peaks between LSK and GMP URE and WT cells? Figure 6E-H should also have the signature number added to the motif analysis.

16. In the RUNX1 ChIP-seq traces in Figure 7D, is RUNX1 not normally bound to these genes? Is this an example of inappropriate RUNX1 binding? Additionally, does PU.1 also bind these sites in AML or are these RUNX1-only bound regions since RUNX1 is a known binding partner of PU.1?

17. Based on data presented in Figs. 6-7, does knockdown of RUNX1 result in collapse of signature 3? Do URE cells have a unique dependence on RUNX1 for their survival as compared to WT cells?

18. Overall, the figure titles are vague and should provide more insight than just the experimental approached.

19. S2C photomicrographs are poor quality, cell and nuclear morphologies are not clear.

Referee #3:

Mice in which a major PU.1 (Spi1) enhancer (URE) is deleted have already been extremely illuminating for the field. This paper takes the phenotype of the PU.1 deltaURE mouse to a depth of investigation that provides repeated surprises and insights, extending beyond the initial model system, and making nearly all points with strong support. It is exciting and filled with novel insights, not only about the role of PU.1 in hematopoietic progenitors but also about the control of a specific set of autophagy regulators that appear to support the preleukemic state. The authors identify a novel set of PU.1-inhibited target genes that they show to be implicated functionally in preleukemic and leukemic cell survival as well as in autophagy, arguing that their upregulation in the URE-deleted, PU.1-deficient mice enables preleukemic cells to gain an advantage. They find evidence that a substantial panel of human AML samples likewise overexpress these genes. The authors finally show that these autophagy regulators are among a set of genes that are stimulated primarily by Runx factors that are released from interaction with PU.1 when PU.1 levels are reduced. These results thus provide strong support for the idea that Runx1 has different functions when binding with PU.1 than when it is in excess over PU.1.

The biology revealed thus prompts the reader to look at non-oncogenic fate decisions as well as preleukemia in a new and different way. Still, a few points in the manuscript could be clarified further for the readers.

1. The authors show a UMAP plot of the integrated data from wildtype and URE-deleted single-cell transcriptomes from Lin- Kit+ hematopoietic precursors. An important part of their focus is on cluster 1, the most stem/progenitor-like. However, it would be very useful to show separately the cells from wildtype and URE-deleted mice in a split plot, so that any shifts in distribution could be seen. Fig. 1E shows that in fact the genes expressed by the two genotypes are not the same even within the same cluster. It would be good to see directly how much the overall pattern is changed.

2. In Fig. 6, the authors provide a good argument based on normal PU.1 binding patterns that signatures 1 & 2 genes are directly positively regulated by PU.1, while signatures 3 & 4 genes are indirectly repressed by PU.1, probably through effects on Runx. But the genes in Signature 3 would offer a long-term selective advantage for cells in the mutant, and their increased expression in murine Lin- Kit+ cells and long-term cell lines might partly reflect a slow population selection phenomenon. Is there a model system in which the authors could delete the URE experimentally in a normal hematopoietic population at a particular time, then show how quickly and how uniformly the changes in the expression of these different gene sets occur? This might not be feasible, but it would add strongly to the evidence that PU.1 was the primary repressor keeping the signature 3 and 4 genes downregulated.

3. The discovery that PU.1 represses key autophagy regulators seems new and intriguing. However, several points are not clear. Does this fit with the phenotype that Staber et al. 2013 described previously, with PU.1 actively inhibiting proliferation in the stem cell compartment? Was a difference seen, especially in cluster 1, between the cell cycle gene expression of the WT and URE-deleted populations? Also, while the single-cell UMAP plot in Fig. 1 apparently came from an analysis that had cell cycle-related genes removed, it seems that the relationship between these two possible effects of PU.1 would be easier to see if some of the data could be presented with the cell cycle genes included in the analysis, to show whether the Signature 3 gene-increased cells were the same as the ones showing increased proliferation or not.

4. Minor comments:

a. The supplementary tables do not have their names associated with the individual files. It is not clear which one is which except by opening each file.

b. On p. 4 of the supplementary data, referring to the "lessAbs" test in DESeq2, the text says that non-differentially accessible regions would be selected with an "FDR < 0.05". It seems this should be FDR > 0.05.

c. A number of the figures would be clearer with more labeling. In Figs. 2C, D and 6A-D, directions of comparisons are labeled "up" and "down" but do not indicate whether "up" is up in URE-deleted or up in wildtype. Supplementary Fig. 1B-E do not

indicate which plots are from before and which are from after applying filters. In Supplementary Fig. 2, panel A does not label VCP or PU.1, and panel D shows a ranking but does not indicate which genotype is in the lower numbers and which in the higher ones ("vs" does not necessarily mean "left to right"). The same issue in Supplementary Figure 3B, where it is not clear which genotype values are being divided by which in the fold change. This information could be in the legends but it should be somewhere associated with the figures.

d. In Fig. 7C, the signature 4 color is so light that it becomes invisible.

5. Nonessential but of interest: The transcription factor motifs shown in Fig. 6G are reminiscent of motifs seen in lymphocytes. Is there any evidence that the progenitors in the URE-deleted mice are poised in a lymphoid-biased state?

Bender et al.

Title: Redistribution of PU.1 partner transcription factor binding secures cell survival in leukemogenesis

EMBOJ-2024-116753

Detailed response to reviewers' criticisms to the authors:

Reviewer #1:

General reviewer comment: Overall, the experiments are well thought and convincing, many of the findings are quite novel and the paper is well-written, although the Introduction needs to be expanded a little bit to better introduce the reader into the topic. Below are few suggestions to help the authors prepare a revised version of their manuscript.

Response: We thank the reviewer for the positive evaluation of our manuscript. According to the reviewer's suggestion, we have expanded the introduction on pages 2-4.

Point 1: In figure 1B the authors should include a heatmap depicting the expression levels of marker genes in individual cells in addition to the provided averaged heatmap, so that readers can better appreciate the expression levels and the percentage of cells in which each marker is expressed. The same should be done for Figure 2A.

Response to point 1: We agree that the per-cluster expression levels in the scRNA-seq datasets should be displayed more transparently. We have slightly deviated from the reviewer's suggestion of a heatmap, as heatmaps in our experience tend to be difficult to examine on a per-gene basis when displaying hundreds of genes for thousands of single cells. Instead, we added Figure S1F, which contains boxplots showing the percentage of expression per cluster toward the per-cluster marker genes. The minimum percentage here is 25%, which is in line with our statement in the methods section that only genes with >25% cluster expression were eligible for marker detection. This cutoff was partly based on work from Sonesson et al. (<https://www.nature.com/articles/nmeth.4612>) highlighting the importance of such prefiltering to avoid excessive false positives. Furthermore, we added Figure S1G, which shows a dot plot highlighting relative expression and percentage of expression for canonical marker genes per cluster, matching the highlighted genes from Figure 1B (those displayed left/right of the heatmap). We also added to the legend of Figure S1 a sentence referring the reader to Supplemental Table S1, which contains the percentage of expression per cluster for all marker genes.

We also added Figure 2B, containing boxplots that indicate the percentage of expression for the signatures from Figure 2A.

Point 2: In Figure 1C the authors could add a dot plot depicting the expression of PU.1 in each individual cell as arranged in the UMAP, in addition to the already provided boxplots.

Response to point 2: According to the reviewer's suggestion, we added Figure S1I showing a UMAP split by genotype and colored by PU.1 expression levels to better visualize the differential expression pattern for this gene and its general downregulation in URE cells.

Point 3: In figure 6(E-H) the percentage of regions in which each motif is found can be added, together with the respective abundance in background regions, so that readers can appreciate its enrichment in tested regions.

Response to point 3: We added to this figure the requested percentages of motifs found in the testing (foreground) and background sets to more transparently visualize their enrichment.

Point 4: We believe that the available chip-seq and epigenomics data have not been analyzed and visualized in a combinatorial fashion to reach their full potential. Through, the use of heatmaps the authors could better illustrate the redistribution of RUNX1 peaks and what epigenomic markers and other TFs from the literature are co-enriched in these differentially bound sites in order to get a stronger sense of their regulatory potential.

Response to point 4: We addressed this point by integrating our RUNX1 ChIP-seq data from THP-1 cells (with either PU.1 knockdown or non-targeting controls) with several published epigenomics datasets, complemented by a motif enrichment analysis. In the new Figure 7C, we performed motif enrichment analysis contrasting the two sets of genomic loci to suggest context-specific RUNX1 binding partners. In Figure 7D we now show heatmaps highlighting the abundances of published PU.1 ChIP-seq with our own RUNX1 ChIP-seq, published H3K27ac and published ATAC-seq data (published data sources are provided in the "Data usage and availability" section at the end of the manuscript). For this figure, we separated genomic loci from our ChIP-seq dataset into those showing increased or decreased RUNX1 binding in THP-1 upon PU.1 knockdown. As outlined in the revised results section, we highlight that regions that gain RUNX1 binding upon PU.1 knockdown generally have lower chromatin accessibility and activating histone marks, suggesting that these regions harbor the potential to be more activated upon a stimulus, which in our model system is triggered by the downregulation of PU.1.

Point 5: The authors should add a detailed table containing quality metrics for ATAC-seq and ChIP-seq experiments.

Response to point 5: We have added detailed QC metrics for our ATAC-seq and ChIP-seq data in Supplemental Table S7.

Reviewer #2:

General comment: Major criticisms pertain to the very unclear methods descriptions.

Response: We have revised the results and methods sections to be clearer and more transparent regarding how experiments were performed and how data were analyzed. This will be addressed in more detail in the points below.

Point 1: The conclusion from Figure 2 that "self-renewal and differentiation were interconnected" on the basis that the URE cells had decreased stem cell and myeloid genes by scRNA-seq analyses, is not fully supported. Would, for example, rescue of stem cell genes in the URE cells result in the upregulation of the same myeloid genes? Overexpression and/or knockdown experiments would greater support this claim. Why is C5 included in Figure 2A? It is not discussed.

Response to point 1: We apologize that there was obviously a misunderstanding on our conclusion regarding the link between the deregulated self-renewal and differentiation programs in URE^A cells. It was not our intention to claim that these two programs were functionally interconnected. Instead, we meant to conclude that there was a regulatory

interconnection based on their simultaneous transcriptional control by PU.1 in the same single cells. We have now made this point clear on page 7.

Nevertheless, we are intrigued by the reviewer's thought that there may be a functional connection between the PU.1-controlled differentiation and stem cell programs. We thank the reviewer for the excellent suggestion of testing such a functional connection, which we did by assessing the effect of re-expression of stem cell genes toward rescuing the myeloid phenotype of URE^Δ cells. Because URE^Δ cells express greatly decreased levels of the stem and early progenitor cell gene *Flt3*, we transduced Hox-URE cells with a pMIG-*Flt3* overexpression construct (or empty vector as control) and then cultured cells in the presence of an *Flt3* ligand followed by RNA-seq. Indeed, we found that a great number of key genes related to the monomyeloid fate and innate immunity were overexpressed in the *Flt3*-transduced cells compared to controls. Below we show an MA plot contrasting *Flt3* with empty vector-transduced cells, with examples of significant genes related to monomyeloid fate/innate immunity highlighted by name. However, these results are still too preliminary to include into our manuscript, as they require much more robust evaluation and rigorous functional testing. Therefore, we decided to show the results as a reviewer-only figure, but we will follow up this interesting path in the future.

Reviewer-only figure: MA plot (average expression across samples on the x-axis, log₂ fold change on the y-axis) from differential expression analysis. Hox-URE cells were retrovirally transduced with either pMIG-*Flt3* or a pMIG empty vector overexpression construct, and then cultured in the presence of an *Flt3* ligand. RNA-seq was performed to assay the expression status of genes related to myeloid fate and innate immunity. Analysis was conducted with DESeq2. As cutoffs, we used an FDR < 0.05 and absolute fold change beyond 1.5. Genes in red indicate overexpression in pMIG-*Flt3*, while genes in blue indicate overexpression in the empty vector control. Grey genes are not significant. Those genes highlighted by name are significant under these cutoffs and are related to myeloid fate or innate immunity based on manual curation.

Our addition of C5 in Figure 2A was indeed not explained transparently. We now mention in the results section (paragraph starting with “Hierarchical clustering of these genes in cluster C1...” on page 5) that C5 was included to capture monocyte differentiation genes already primed in early HSPCs.

Point 2: Is *Bcl2* the only anti-apoptotic gene that was upregulated in URE compared to WT in signature 3 in Figure 2A? It is postulated that the upregulation of *Bcl2* seen in signature 3 and

identified as an "essential" gene in the shRNA screen in URE cells may help these cells avoid apoptosis that may result from excessive autophagy. It would be beneficial to test whether inhibiting Bcl2 either with genetic approaches or pharmacological inhibitors (like Venetoclax) causes a reduction in autophagy to further support this potential role of Bcl2 balancing the "beneficial and detrimental impacts of autophagy on cell survival".

Response to point 2: We thank the reviewer for encouraging a more detailed assessment of the role of Bcl2 in our model system. In total there were eight genes present in signature 3 linked to "negative regulation of apoptotic signaling pathway" (GO term 2001234), namely Ar, Bcl2, Ctnn, Eya2, Ppif, Rrm2b, Sh3glb1 and Tgfbr1. The total GO term harbored 232 genes. Due to the low coverage of this GO term, and since it did not reach statistical significance in the conducted functional enrichment analysis, we did not report this overlap in the results but concentrated on Bcl2. As suggested by the reviewer, we have now performed an additional experiment by treating both Hox-WT and Hox-URE cells with Venetoclax at two different concentrations to assess survival. We assessed survival rather than autophagy because the ability of Bcl2 to inhibit autophagy is already well established (for example, reviewed in <https://www.ncbi.nlm.nih.gov/pmc/articles/PMC3304572/>). By focusing on survival as readout, we aimed to investigate whether Hox-URE cells were dependent on Bcl2 to block autophagy-associated cell death. As now shown in the new Figure 5F, URE^Δ cells indeed are dependent on Bcl2 to ensure their survival, whereas Bcl2 inhibition in WT cells had only minor effects. This result provided clear evidence for a particular role of Bcl2 in URE^Δ cells. Moreover, this result also provided general validation of our shRNA library screen, as it confirms a differential signature 3 gene requirement for URE^Δ versus WT cells. In the results section, we now describe this experiment on page 14 in the paragraph starting "PU.1-downregulated cells depend on Bcl2 upregulation".

Point 3: The shRNA screen, rationale, analysis plan, and data interpretation are incomplete or unclear. How was 10 cell divisions determined for each cell line? Why were only 3 positive control genes selected for and what is the evidence they are essential genes? Results text indicates shRNA pool was transfected. This is inaccurate, please revise. What was the MOI/infection efficiency of the pooled lentiviral library in each of the cell lines and what was the library coverage in the screen for each timepoint?

Response to point 3: We agree that in our paper's original version, the methods were not described in sufficient depth. Hence, we have revised in particular the supplemental methods (as well as the main text, please see points discussed below) to provide a better description of the experimental aims, procedures and analysis. We now present all experimental details on the shRNA screen, including information on determination of cell divisions, choice of positive controls, target MOI and library coverage in the supplemental methods on pages 5-7 of the supplement, in the section "shRNA screen."

Point 4: Results explain this screen readout as "knockdown" effect which is incorrect. Barcodes are measured and their abundance in the population corresponds to presumed effects of the attached shRNA on the cell its in given its change in abundance compared to input. Revise.

Response to point 4: We have revised the manuscript text according to the reviewer's suggestion on page 8.

Point 5: The description of the shRNA analyses is very confusing in both the results and methods text. The phrasing "more essential" is confusing (Fig. 3C) and how these genes are different than the essential genes in Fig. 3B that are unique to AML and URE is not clear. In Fig. 3B, there are 28 depleted shRNA unique to AML, 18 depleted shRNA unique to URE, and 18 depleted shRNA in common between AML and URE - and these are not overlapping with WT. Why then is the added step of differential fold changes required? In this analysis, were there no uniquely essential genes per any of the three cell lines? It is also confusing that "essential" is inconsistently used in quotations.

Response to point 5: We apologize for causing confusion by our suboptimal choice of words "more essential" and for not clearly describing the analysis rationale. We have revised the results section and methods accordingly, now avoiding the term "more essential."

In detail: The analysis from Figure 3B aimed to identify essential genes per cell line by using a fixed statistical cutoff (FDR < 0.05) based on a permutation test and a minimum number of shRNAs (3 out of 5) to support the classification, as described in the methods. However, in such a cutoff analysis, a gene could be slightly below cutoff in one cell line but slightly above cutoff in another cell line. Therefore, strict overlaps are not sufficient to identify genes with a preference for one over another condition in terms of growth inhibition preferences, which is why we do not use Figure 3B beyond reporting and summarizing the per-cell line results. To overcome this, and to identify those genes that inhibit growth more in URE^Δ compared to WT cells (while still showing significant growth inhibition in both), we used the differential fold change strategy, inspired by a published workflow for high-throughput CRISPR-Cas9 screens (an almost identical analysis to that of shRNA screens) by the authors of the statistical framework (MAGeCK) that we used for the initial definition of essential genes (<https://www.nature.com/articles/s41596-018-0113-7>). This directly compared the normalized fold changes from the individual shRNAs and aimed to quantitatively (rather than by a qualitative overlap) identify genes that upon knockdown cause *preferential growth inhibition* (we now use this term throughout the text instead of "more essential") in URE^Δ versus WT. We hope that now the rationale and analysis is clearer, and we thank the reviewer for bringing this important point to our attention.

Point 6: What is the scientific justification for terming a gene "more essential" in AML or URE cells versus WT? What is the functional and/or biological proof that a gene is more essential to one cell type versus another? How is this cut-off determined and measured for biological significance?

Response to point 6: We hope that our response to point 5 and the associated changes in the text have resolved this issue.

Point 7: Fbxl8, Vmp1, and Acss2 were identified in the focused shRNA screen to be more essential for both URE and URE-AML cells compared to WT. In Figure 4D, of the 3 genes, Vmp1, is the only gene that visually appears to have an effect on survival similar to the killing control. Additionally, the percent survival graph of Acss2 doesn't appear very different compared to NTC. Was there any statistical testing performed in Figure 3D? If so, it would be valuable to include this to demonstrate a significant decrease in cell survival with each shRNA.

Response to point 7: Following the reviewer's suggestions, we have extended our validations of the shRNA results. In particular, we added appropriate statistics (2-way ANOVA to test whether survival decay over time is stronger in target genes compared to NTC) to the validation

experiment in URE-AML cells (now Figure S3F). This shows that impaired cell survival by shRNA-mediated knockdown was statistically significant for each of the three validated genes (Fbxl8, Vmp1 and Acss2). However, while the validation for Fbxl8 and Vmp1 showed strongly reduced survival effects (matching those of the NTC), that of Acss2 had much milder effects (although being statistically significantly different from the NTC), confirming the reviewer's assumption. Therefore, we decided to remove this gene from further discussion and instead focused more on validating Fbxl8 and Vmp1. Accordingly, as suggested by the reviewer, we added new experiments validating the preferential growth inhibition of Vmp1 and Fbxl8 in URE^Δ versus WT cells, which will be discussed in detail in our response to point 10.

Point 8: One of the major claims from the study suggests that autophagy is a pro-oncogenic process in non-transformed cells. Would inhibiting or impairing any of the autophagy factors identified in this work, such as Vmp1, prevent leukemic transformation in the Pu.1 URE mouse model? Does inhibiting Vmp1 alter autophagy in URE-specific cell lines? At present the relationship between PU.1 and autophagy descriptive.

Response to point 8: We thank the reviewer for raising this issue. We have experimentally addressed this point with two independent strategies. First, presented in Figure 5E, we transduced both Hox-WT and Hox-URE cells with an anti-Vmp1 shRNA and then measured autophagic flux by CytoID. This indeed confirmed the notable dependency of URE cells on Vmp1, while effects in WT cells were modest. Next, presented in Figure 5H, we re-expressed PU.1 in Hox-URE cells using an estrogen-inducible retroviral overexpression construct, and assessed autophagy by Cyto-ID which was significantly reduced upon re-expression. We describe this in the last paragraph on page 14.

Point 9: The sentence "FBXL8, VMP1 and ACSS2, which were identified as "essential" in both Hox-URE and URE-AML but not Hox-WT cells in our shRNA screen" contradicts the finding from Figure 3 that claims that these three genes were considered "more essential for both Hox-URE and URE-AML than Hox-WT", implying that these genes are still essential in the Hox-WT cells. What was the need for calculating differential fold changes between AML and URE-Hox vs WT?

Response to point 9: As eluded in response to point 5, we agree that our initial description of the shRNA effects in which we used the terms *essential* and *more essential* was confusing. We hope that our changes in the text (please see points 5 and 6) have now resolved this, as we now emphasize that we calculated the differential fold changes for genes essential to either Hox-URE or URE-AML and that the intended readout and narrative is to enrich for genes that show a preferential (quantitative rather than by a qualitative overlap analysis) growth inhibition compared to WT.

Point 10: A more robust validation of more essential genes in AML and URE-Hox would be to use independent shRNA in all 3 cell lines, validate shRNA target repression in all 3 cell lines, and demonstrate that WT-Hox have unaffected growth by knockdown of FBXL8, VMP1, and ACSS2 as compared to URE-Hox and AML which are expected to have growth retardation.

Response to point 10: We have performed an additional validation of the shRNA screen addressing preferential growth inhibition of certain genes in Hox-URE compared to Hox-WT cells (see our response to point 7). For this, as now shown in revised Figure 3D, we transduced Hox cell lines with two independent shRNAs against Vmp1 or Fbxl8 and measured survival after four days. Hox-URE cells show a statistically significant (one-way ANOVA) greater

reduction of growth compared to WT cells for both tested shRNAs, highlighting the preferential growth inhibition of these genes in URE^Δ cells. Moreover, as described in our response to point 2, Bcl2 inhibition by venetoclax has also confirmed a much greater growth retardation effect on URE^Δ than WT cells, providing validation via a different method (chemical inhibition as opposed to shRNA knockdown).

Point 11: The claim that human AML "preferentially overexpress PU.1-repressed genes that were required for both preleukemic and leukemic phases in the URE^Δ model" is not convincing for all of the genes shown in Figure 4A. For example, for FBXL8, 10/13 AML subtypes have a DEG of 0.

Response to point 11: We apologize for the misunderstanding, as we meant that human AML "preferentially" expresses those genes *within* the signature 3 gene cohort that were required for both preleukemic and leukemic URE^Δ cells rather than other signature 3 genes. We did not intend to suggest that all AMLs overexpress these genes. We have modified the text on page 11 to be more accurate here.

Point 12: It is unclear in Figure 4C what the gene expression changes are relative to in all of the gene expression plots, except for PU.1. The claim that PU.1 knockdown lead to reduced expression in the genes (pg.10-11) listed in Figure 4C is also not supported because the expression of ACSS2, C15orf39, FBXL8, PLEKHA5, RAB37, and VMP1 are all higher in the PU.1 KD compared to NTC.

Response to point 12: We thank the reviewer for pointing out that we mixed up the terms *reduced* and *enhanced* here. The correct claim is that PU.1 knockdown leads to enhanced expression in the genes. We have changed the text on page 11 accordingly. We have also added a sentence to the legend of Figure 4C explaining that a relative expression of 1 represents the sample with highest expression, upon which all other values were scaled proportionally.

Point 13: The western blot in Figure 5C does not have the lanes labeled. It also unclear which bands were used for the normalized intensity since there are two LC3B bands. Is VCP used as a loading control?

Response to point 13: We apologize for the missing information. We have added boxes to Figure 5C to highlight which bands were used for quantification. The legend was revised accordingly to indicate what these boxes represent and that VCP was indeed the internal control.

Point 14: Flt3 was re-expressed in Hox-URE cells followed by stimulation with Flt3 ligand to impair autophagy. Is this a novel finding or are there any published studies that demonstrate Flt3 signaling having a role in relating autophagy? Suggesting a link between cytokine signaling and autophagy based on these data in Fig. 5E is premature.

Response to point 14: To the best of our knowledge, a direct link has not yet been made between Flt3 in non-transformed cells and autophagy (though many reports show Flt3-ITD is connected to autophagy in leukemic cells), at least according to the existing literature available through PubMed and Google Scholar. Moreover, according to the reviewer's suggestion, we have now indicated on page 14 that this "potential" link between cytokine signaling and autophagy needs further rigorous mechanistic investigation.

Point 15: Figure 6A-D should have the comparisons labeled on each plot as well as LSK and GMP. Are there overlap in ATAC peaks between LSK and GMP URE and WT cells? Figure 6E-H should also have the signature number added to the motif analysis.

Response to point 15: We made the requested changes to Figure 6E-H, adding the comparisons and cell type labels.

Point 16: In the RUNX1 ChIP-seq traces in Figure 7D, is RUNX1 not normally bound to these genes? Is this an example of inappropriate RUNX1 binding? Additionally, does PU.1 also bind these sites in AML or are these RUNX1-only bound regions since RUNX1 is a known binding partner of PU.1?

Response to point 16: Unfortunately, there is (to the best of our knowledge after checking the GEO databases) no public dataset available that contains ChIP-seq data toward RUNX1 in both primary human AML cells and matched normal controls, such as CD34+ cells that had been produced in the same experimental batch using identical antibodies and treatment protocols. Hence, we unfortunately cannot make a statement on whether these sites are normally bound by RUNX1. We strongly recommend against making statements by comparing independent ChIP-seq datasets, as this assay harbors considerable noise and is prone to extensive batch effects (which is why our own ChIP-seq data are produced in balanced batches and with proper experimental replication). We hope that the reviewer agrees that this question currently cannot be answered given the absence of appropriate datasets.

In order to address whether these loci bind PU.1, we made use of PU.1 ChIP-seq data from THP-1 (which we used already in this study in Figure 7); none of the loci shown in Figure 7F indicated any evidence for PU.1 binding. We show below a reviewer-only figure to illustrate this. In the text, we mention on page 19 that no evidence for PU.1 binding could be found for these loci. With Figure 7F, we highlight that RUNX1 binding at these loci indeed exists also in primary human AML.

Reviewer-only figure: Loci in the human genome (as in Figure 7F) with ChIP-seq tracks toward RUNX1 (GSE60130) in human AML blasts and PU.1 (GSE128834) in parental THP-1 cells. Tracks were normalized to reads per million and normalized per locus to the panel with the highest signal. The CSF1R locus was included to demonstrate that the absence of a PU.1 signal in other tracks was not due to general PU.1 depletion or poor immunoprecipitation quality in this experiment.

Point 17: Based on data presented in Figs. 6-7, does knockdown of RUNX1 result in collapse of signature 3? Do URE cells have a unique dependence on RUNX1 for their survival as compared to WT cells?

Response to point 17: Thank you for raising this important question. We have experimentally addressed this point by transducing Hox-WT and Hox-URE cells with a dominant negative Runx (dnRunx) construct. As now shown in the new Figure 7H, Hox-URE cells show a greater growth reduction than Hox-WT across the two time points measured, and this difference is statistically significant (one-way ANOVA). However, as expected from the current literature, WT cells also show a response, which is most likely caused by the fact that dnRunx knocks down the function of all Runx factors. Nevertheless, we chose to use dnRunx rather than knocking down Runx1 specifically, as Runx factors can act redundantly, compensating for each other when knocked down individually by shRNAs (as previously shown, for example, in T cell development: <https://www.pnas.org/doi/full/10.1073/pnas.2019655118>).

Point 18: Overall, the figure titles are vague and should provide more insight than just the experimental approach.

Response to point 18: We have now modified all figure titles according to the reviewer's suggestion.

Point 19: S2C photomicrographs are poor quality, cell and nuclear morphologies are not clear.

Response to point 19: We have replaced the photomicrographs with better images and added scalebars (Figure S2C).

Reviewer #3:

General reviewer comment: These results thus provide strong support for the idea that Runx1 has different functions when binding with PU.1 than when it is in excess over PU.1. The biology revealed thus prompts the reader to look at non-oncogenic fate decisions as well as preleukemia in a new and different way. Still, a few points in the manuscript could be clarified further for the readers.

Response: We appreciate the positive evaluation of our manuscript and are grateful for the constructive suggestions.

Point 1: The authors show a UMAP plot of the integrated data from wildtype and URE-deleted single-cell transcriptomes from Lin- Kit+ hematopoietic precursors. An important part of their focus is on cluster 1, the most stem/progenitor-like. However, it would be very useful to show separately the cells from wildtype and URE-deleted mice in a split plot, so that any shifts in distribution could be seen. Fig. 1E shows that in fact the genes expressed by the two genotypes are not the same even within the same cluster. It would be good to see directly how much the overall pattern is changed.

Response to point 1: We agree with the reviewer that a more transparent visualization of the cellular proportions will help readers appreciate the redistribution caused by the genotype. For this, we have added Figure S1J, a UMAP colored by cluster membership and overlaid with contour lines that indicate the cellular densities, such that readers can appreciate the depletion of URE cells in the monocyte progenitor compartment and the increased number of neutrophil progenitors.

Point 2: In Fig. 6, the authors provide a good argument based on normal PU.1 binding patterns that signatures 1 & 2 genes are directly positively regulated by PU.1, while signatures 3 & 4 genes are indirectly repressed by PU.1, probably through effects on Runx. But the genes in Signature 3 would offer a long-term selective advantage for cells in the mutant, and their increased expression in murine Lin- Kit+ cells and long-term cell lines might partly reflect a slow population selection phenomenon. Is there a model system in which the authors could delete the URE experimentally in a normal hematopoietic population at a particular time, then show how quickly and how uniformly the changes in the expression of these different gene sets occur? This might not be feasible, but it would add strongly to the evidence that PU.1 was the primary repressor keeping the signature 3 and 4 genes downregulated.

Response to point 2: We thank the reviewer for this suggestion. Unfortunately, as the reviewer correctly assumes, there is no model system in which we could conditionally delete the URE at a particular time. However, we have experimentally addressed this interesting point by a different yet similarly appropriate scenario: We retrovirally transduced Hox-URE cells with an estrogen-inducible PU.1-encoding construct to restore PU.1 expression, then we measured autophagy after addition of estrogen to the medium. Parental Hox-URE cells have enhanced

autophagy compared to Hox-WT cells, which, as we can now show, was greatly reduced upon re-expression of PU.1. This rapid PU.1 rescue effect on autophagy (which, as we show in our manuscript, is driven by signature 3 genes) argues against the possibility of a slow population selection effect. We hope that this result (which we present as Figure 5H, and describe on page 14 of the manuscript) sufficiently addresses this reviewer's question.

Point 3: The discovery that PU.1 represses key autophagy regulators seems new and intriguing. However, several points are not clear. Does this fit with the phenotype that Staber et al. 2013 described previously, with PU.1 actively inhibiting proliferation in the stem cell compartment? Was a difference seen, especially in cluster 1, between the cell cycle gene expression of the WT and URE-deleted populations? Also, while the single-cell UMAP plot in Fig. 1 apparently came from an analysis that had cell cycle-related genes removed, it seems that the relationship between these two possible effects of PU.1 would be easier to see if some of the data could be presented with the cell cycle genes included in the analysis, to show whether the Signature 3 gene-increased cells were the same as the ones showing increased proliferation or not.

Response to point 3: We thank the reviewer for raising this important question about comparing our results to those published by Staber et al., which we have addressed by an additional analysis. Staber et al. made the point that in their model of PU.1 hypomorphic mice (which was achieved by ablation of PU.1 auto-regulation through removal of the three PU.1 binding sites with the PU.1 URE), the HSC compartment overexpressed several cell cycle activators, leading to *increased* proliferation and thereby HSC exhaustion. However, in our model (in which PU.1 downregulation by full deletion of the URE was greater than in the Staber model), the progenitor compartment shows a general hypoproliferation (Rosenbauer et al. 2004 Nat Genet). For example, in preparation for our shRNA screen, we extensively assayed Hox-WT and Hox-URE cell lines for the rate of cell division, which was almost double in a given time frame for WT compared to URE cells (data not shown). Staber et al. explained their phenotype as arising from the differential expression of key cycle regulators that are either up- or downregulated (see, for example, Figure 3E in Staber et al.). We evaluated the expression patterns of these genes in our cluster C1, contrasting URE and WT cells, and we also subjected C1 cells to cell cycle phase classification. As shown in a reviewer-only figure below, only one of these genes (*Ccnd1*) reached statistical significance using higher expression in WT cells, while all others did not show significant differences. Also, in the same figure we estimated cell cycle phases of WT and URE cells using the automated "cyclone" classifier (<https://pubmed.ncbi.nlm.nih.gov/26142758/>), indicating that URE^Δ mice have a reduced fraction of G2M cells, suggesting hypoproliferation.

Reviewer-only figure: (Left) Single-cell RNA-seq differential expression analysis contrasting C1 cells of URE with WT cells, focused on a set of cell cycle- and proliferation-related genes as described in Staber et al. (2013) Figure 3E. Positive fold changes indicate higher expression in URE cells. FDRs were calculated with DESeq2 using standard settings. (Right) Automated cell cycle classification of C1 WT and URE cells using the cyclone classifier from the R/Bioconductor scan.

Furthermore, in response to this reviewer's point, we would like to highlight that the absence of cell cycle genes as described in the methods only refers to the choice of genes used for clustering, not for the general analysis of differentially expressed genes. This means that *no relevant cell cycle effects are lost*. For example, signature 1 (genes generally downregulated in early URE^A progenitors) contain the key proliferation regulators Ccnd1/2 and Ki67 (Mki67), despite that these genes were removed for the clustering step. We apologize for the confusion and have now made it clear in the supplemental methods section that the removed cell cycle genes were eligible for all analyses downstream of the clustering step.

Point 4a: The supplementary tables do not have their names associated with the individual files. It is not clear which one is which except by opening each file.

Response to point 4a: We thank the reviewer for bringing this to our attention. We will make sure that upon resubmission, the file names will be correctly associated with the files.

Point 4b: On p. 4 of the supplementary data, referring to the "lessAbs" test in DESeq2, the text says that non-differentially accessible regions would be selected with an "FDR < 0.05". It seems this should be FDR > 0.05.

Response to point 4b: We apologize for this confusion and have clarified this section. In fact, FDR < 0.05 is correct, as this test is specifically testing *against* differential accessibility. Hence, low FDRs indicate strong evidence that a region is not differential or is "the same."

Point 4c: A number of the figures would be clearer with more labeling. In Figs. 2C, D and 6A-D, directions of comparisons are labeled "up" and "down" but do not indicate whether "up" is up in URE-deleted or up in wildtype. Supplementary Fig. 1B-E do not indicate which plots are from before and which are from after applying filters. In Supplementary Fig. 2, panel A does not label VCP or PU.1, and panel D shows a ranking but does not indicate which genotype is in the lower numbers and which in the higher ones ("vs" does not necessarily mean "left to right"). The same issue in Supplementary Figure 3B, where it is not clear which genotype values are being divided by which in the fold change. This information could be in the legends but it should be somewhere associated with the figures.

Response to point 4c: We have made changes to the figures so that readers can clearly appreciate which comparisons were made and whether positive and negative fold changes mean over- or underexpression in WT or URE cells.

Point 4d: In Fig. 7C, the signature 4 color is so light that it becomes invisible.

Response to point 4d: We changed the color in this figure (former Figure 7C, now Figure 7E) to provide better visibility.

Point 5: Nonessential but of interest: The transcription factor motifs shown in Fig. 6G are reminiscent of motifs seen in lymphocytes. Is there any evidence that the progenitors in the URE-deleted mice are poised in a lymphoid-biased state?

Response to point 5: This is indeed a very interesting point. To address it, we have used the scRNA-seq data for C1 (early progenitor) cells and scored these cells with two lymphoid genesets, namely (GO: 0030183) B cell differentiation and (GO: 0030217) T cell differentiation. Below we added a reviewer-only figure showing boxplots for these genesets colored by genotype; yet, this analysis failed to identify a conclusive lymphoid bias by genotype.

Reviewer-only figures: Cells of cluster C1 were scored with the R/Bioconductor UCell towards two lymphoid genesets obtained from Gene Ontology (GO: 0030217 – T Cell Differentiation, GO: 0030183 – B Cell Differentiation). Boxplots indicate per-cell geneset scores.

Dear Dr Rosenbauer,

Thank you for submitting a revised version of your manuscript. Your study has now been seen by all original referees, who find that their previous concerns have been addressed and now recommend publication of the manuscript. There remain only a few mainly editorial points that have to be addressed before I can extend formal acceptance of the manuscript:

- Please double-check to make sure to all relevant funding information in the manuscript is also entered into our submission system. (Missing in the system currently: the Interdisciplinary Center of Clinical Research (IZKF grant Ros4-003-23))
- Please rename the "Data usage and availability" section to "Data Availability"
- Please rename the Conflict of Interest section into "Disclosure and Competing Interests Statement", in accordance with our updated Guide to Authors (<https://www.embopress.org/competing-interests>)
- CRediT has replaced the traditional author contributions section because it offers a systematic machine readable author contributions format that allows for more effective research assessment. Please remove the Author Contributions section from the manuscript and use the free text boxes beneath each contributing author's name in our online systems to add specific details on the author's contribution. More information is available in our guide to authors.
- There is a reference to "data not shown" on page 19. According to our policy, which does not permit references to "data not shown", please include this information in the Appendix. Please see also <https://www.embopress.org/page/journal/14602075/authorguide#unpublisheddata>.
- Please provide either a "Yes" or a "Not Applicable" answer to each one of the questions in your Author Checklist (https://www.google.com/url?sa=t&source=web&rct=j&opi=89978449&url=https://www.embopress.org/pb-assets/embo-site/EMBO%2520Press%2520Author%2520Checklist-1642513524327.xlsx&ved=2ahUKEwiltPPLrJOJAxWQ_rsIHae2JWQQFnoECBUQAQ&usg=AOvVaw1sqIXyQPtP8HrZzFpAYtut) as you have not answered any of them yet. In the last column of this checklist, only the sections of the manuscript where the relevant information can be found should be listed (the information per se should be included in the main manuscript file).
- Please upload figures as individual high-resolution figure files; figure legends should be included in ms file
- Please rename and upload Supplemental Tables 1, 3, 4, 6, 7, 8 and 9 as Datasets: Dataset EV1-EV7 with the legends uploaded as a separate tab in each Excel file and corresponding callouts; Supplemental Tables 2 and 5 should be renamed and uploaded as EV tables: Table EV1-EV2 with the corresponding callouts
- Please convert the Appendix file in PDF format. The nomenclature should be Appendix Figure Sx with the appropriate callouts; Please also add page numbers in ToC
- Please provide a resource and tools table (https://www.embopress.org/pb%2Dassets/embo-site/Reagents_Tools_Table_TEMPLATE.docx)
- Please save the Source data files in a scheme one figure/folder and then uploaded as .zip files. E.g. all the Source data files for figure 1 need to be saved in a single folder and this needs to be zipped and then uploaded as "SD figure 1.zip" file. For EV and/or appendix figures, ZIP together all source data.
- Please provide suggestions for a short 'blurb' text prefacing and summing up the conceptual aspect of the study in two sentences (max. 250 characters), followed by 3-5 one-sentence 'bullet points' with brief factual statements of key results of the paper; they will form the basis of an editor-written 'Synopsis' accompanying the online version of the article. Please also provide an altered synopsis image, making sure that the aspect ratio conforms to our website's format - it should be exactly 550 pixels wide and between 300-600 pixels high.
- Please provide the specific URL for GSE251674 dataset in the data availability statement.
- Please define the box plots in terms of minima, maxima, centre, bounds of box and whiskers, and percentile in the legends of figures 1c; 2b; 4b-c; 5e, g.
- Please provide the information related to n in the legends of figures 1c; 2b, d; 4b; 7a.
- Although 'n' is provided, please describe the nature of entity for 'n' in the legends of figures 4c; 5c-f.

- Please define the error bars in the legends of figures 5f; 7a.

- Please modify the session order which should be: title page with complete author information, abstract, keywords, introduction, results, discussion, methods, data availability section, acknowledgements, disclosure and competing interests statement, references, main figure legends, tables, expanded figure legends.

With best regards,

Cornelius Schneider

Cornelius Schneider, PhD
Editor | The EMBO Journal
c.schneider@embojournal.org

Referee #2:

In this revised manuscript the authors adequately addressed all of the concerns raised in the prior review. The text is written more clearly and the manuscript overall is significantly improved.

Referee #3:

This is an exciting paper with far-reaching implications. The authors have done a great job of responding to the reviews. The added detail not only clarifies the experiments but adds more richness to the implications of the data. I have no further issues, and I thank the authors for their thoughtful responses.

Bender et al.

Title: Redistribution of PU.1 partner transcription factor binding secures cell survival in leukemogenesis

EMBOJ-2024-116753R

Point-by-point response to remaining issues:

Point 1: Please double-check to make sure to all relevant funding information in the manuscript is also entered into our submission system. (Missing in the system currently: the Interdisciplinary Center of Clinical Research (IZKF grant Ros4-003-23))

Response to point 1: All relevant funding information is now entered into the submission system.

Point 2: Please rename the "Data usage and availability" section to "Data Availability"

Response to point 2: We renamed as requested.

Point 3: Please rename the Conflict of Interest section into "Disclosure and Competing Interests Statement", in accordance with our updated Guide to Authors (<https://www.embopress.org/competing-interests>)

Response to point 3: We renamed as requested.

Point 4: CRediT has replaced the traditional author contributions section because it offers a systematic machine readable author contributions format that allows for more effective research assessment. Please remove the Author Contributions section from the manuscript and use the free text boxes beneath each contributing author's name in our online systems to add specific details on the author's contribution. More information is available in our guide to authors.

Response to point 4: We removed the section as requested and used the online system instead.

Point 5: There is a reference to "data not shown" on page 19. According to our policy, which does not permit references to "data not shown", please include this information in the Appendix. Please see

also <https://www.embopress.org/page/journal/14602075/authorguide#unpublisheddata>.

Response to point 5: We added as Appendix Fig. 5C a genomic browser track that highlight the statement that PU.1 does not bind to the genomic loci of discussed genes. In fact, it is the same panel we included as "reviewer-only" in response to reviewer 2 - point 16. Hence, the reviewer has seen and approved it.

Point 6: Please provide either a "Yes" or a "Not Applicable" answer to each one of the questions in your Author Checklist

(<https://www.google.com/url?sa=t&source=web&rct=j&opi=89978449&url=https://www.embo>

Bender et al.

press.org/pb-assets/embo-site/EMBO%2520Press%2520Author%2520Checklist-1642513524327.xlsx&ved=2ahUKEwiltPPLrJOJAxWQ_rslHae2JWQQFnoECBUQAQ&usg=AOvVaw1sqlXyQPtP8HrZzFpAYtut) as you have not answered any of them yet. In the last column of this checklist, only the sections of the manuscript where the relevant information can be found should be listed (the information per se should be included in the main manuscript file).

Response to point 6: We have filled out the checklist as requested.

Point 7: Please upload figures as individual high-resolution figure files; figure legends should be included in ms file

Response to point 7: We have uploaded the main figures as high-resolution TIF files and provide the legends as a separate ms document.

Point 8: Please rename and upload Supplemental Tables 1, 3, 4, 6, 7, 8 and 9 as Datasets: Dataset EV1-EV7 with the legends uploaded as a separate tab in each Excel file and corresponding callouts; Supplemental Tables 2 and 5 should be renamed and uploaded as EV tables: Table EV1-EV2 with the corresponding callouts

Response to point 8: We have renamed these files as requested and made the corresponding changes in both the main text and appendix.

Point 9: Please convert the Appendix file in PDF format. The nomenclature should be Appendix Figure Sx with the appropriate callouts; Please also add page numbers in ToC

Response to point 9: We provide the Appendix as PDF and made the requested changes.

Point 10: Please provide a resource and tools table

(https://www.embopress.org/pb%2Dassets/embo-site/Reagents_Tools_Table_TEMPLATE.docx)

Response to point 10: We filled out the template sheet with the relevant information.

Point 11: Please save the Source data files in a scheme one figure/folder and then uploaded as .zip files. E.g. all the Source data files for figure 1 need to be saved in a single folder and this needs to be zipped and then uploaded as "SD figure 1.zip" file. For EV and/or appendix figures, ZIP together all source data.

Response to point 11: We have adopted this folder scheme for all source data.

Point 12: Please provide suggestions for a short 'blurb' text prefacing and summing up the conceptual aspect of the study in two sentences (max. 250 characters), followed by 3-5 one-sentence 'bullet points' with brief factual statements of key results of the paper; they will form the basis of an editor-written 'Synopsis' accompanying the online version of the article.

Bender et al.

Please also provide an altered synopsis image, making sure that the aspect ratio conforms to our website's format - it should be exactly 550 pixels wide and between 300-600 pixels high.

Response to point 12: We have uploaded suggestions for a 'blurb' text and a graphical abstract.

Point 13: Please provide the specific URL for GSE251674 dataset in the data availability statement.

Response to point 13: We have added a URL as requested.

Point 14: Please define the box plots in terms of minima, maxima, centre, bounds of box and whiskers, and percentile in the legends of figures 1c; 2b; 4b-c; 5e, g.

Response to point 14: We added the definitions to the respective legends.

Point 15: Please provide the information related to n in the legends of figures 1c; 2b, d; 4b; 7a.

Response to point 15: We have added the requested information into the respective figure legends.

Point 16: Although 'n' is provided, please describe the nature of entity for 'n' in the legends of figures 4c; 5c-f.

Response to point 16: We have added the requested information into the respective figure legends.

Point 17: Please define the error bars in the legends of figures 5f; 7a.

Response to point 17: We have defined the error bars in figure 5f and the horizontal line in figure 7a.

Point 18: Please modify the session order which should be: title page with complete author information, abstract, keywords, introduction, results, discussion, methods, data availability section, acknowledgements, disclosure and competing interests statement, references, main figure legends, tables, expanded figure legends.

Response to point 18: We reorder the session as requested.

Dear Prof. Rosenbauer,

I am pleased to inform you that your manuscript has been accepted for publication in the EMBO Journal.

Yours sincerely,

Cornelius Schneider, PhD
Editor
The EMBO Journal
c.schneider@embojournal.org
